EMBO
Molecular Medicine

# Prolonged contact with dendritic cells turns lymph node-resident NK cells into anti-tumor effectors

Francesca Mingozzi[1], Roberto Spreafico[1], Tatiana Gorletta[1], Clara Cigni[1], Marco Di Gioia[2], Michele Caccia[3], Laura Sironi[3], Maddalena Collini[3], Matias Soncini[4], Michela Rusconi[1], Ulrich H von Andrian[5,6], Giuseppe Chirico[3], Ivan Zanoni[1,2,4,*,†] & Francesca Granucci[1,4,**,†]

## Abstract

Natural killer (NK) cells are critical players against tumors. The outcome of anti-tumor vaccination protocols depends on the efficiency of NK-cell activation, and efforts are constantly made to manipulate them for immunotherapeutic approaches. Thus, a better understanding of NK-cell activation dynamics is needed. NK-cell interactions with accessory cells and trafficking between secondary lymphoid organs and tumoral tissues remain poorly characterized. Here, we show that upon triggering innate immunity with lipopolysaccharide (LPS), NK cells are transiently activated, leave the lymph node, and infiltrate the tumor, delaying its growth. Interestingly, NK cells are not actively recruited at the draining lymph node early after LPS administration, but continue their regular homeostatic turnover. Therefore, NK cells resident in the lymph node at the time of LPS administration become activated and exert anti-tumor functions. NK-cell activation correlates with the establishment of prolonged interactions with dendritic cells (DCs) in lymph nodes, as observed by two-photon microscopy. Close DC and NK-cell contacts are essential for the localized delivery of DC-derived IL-18 to NK cells, a strict requirement in NK-cell activation.

**Keywords** dendritic cells; immunosurveillance; innate immunity; natural killer cells; two-photon microscopy

**Subject Categories** Cancer; Immunology

## Introduction

Natural killer (NK) cells are lymphocytes endowed with germ-line-encoded receptors that do not undergo somatic rearrangements. Due to the absence of clonally distributed receptors, NK cells are classified as innate lymphocytes. NK cells are among the first lymphocytes involved in sensing host cell modifications due to infections or to tumor transformation (Moretta *et al*, 2014). In several types of inflammatory processes, NK lymphocytes are the earliest source of pro-inflammatory cytokines, particularly IFN-γ, which, in turn, potently activates macrophages, initiating type I innate (M1) and adaptive (Th1) immunity (Martin-Fontecha *et al*, 2004; Agaugue *et al*, 2008; Kupz *et al*, 2013; Lee *et al*, 2015).

Activation of type I immunity is essential not only to adequately respond to viral or bacterial infections but also for tumor immuno-surveillance. Accordingly, low levels of NK-cell activity usually correlate with a high risk to develop cancer, and high numbers of NK cells in the tumor microenvironment correlate with improved prognosis of tumor-affected patients (Desbois *et al*, 2012).

Understanding the mechanisms and the dynamics of NK-cell responses, including the timing and site of NK-cell activation, is therefore important to improve anti-tumor immunotherapies. A common feature of tumor growth is the capacity of tumor cells to create an immunosuppressive microenvironment. Therapeutic strategies aimed at generating appropriate inflammatory conditions to reactivate immunosurveillance are based on cytokine administration or vaccination, as well as treatments with Gram-negative and Gram-positive bacteria (Toussaint *et al*, 2013; Lin *et al*, 2015; Thompson *et al*, 2015). Considering the potent pro-inflammatory activity of NK cells, it would be desirable to design vaccination protocols that strongly boost NK-cell activation.

NK-cell activation depends on the balance between activating and inhibitory signals transduced by germ-line-encoded receptors. Over the last few decades, it has been clearly demonstrated that

1   Department of Biotechnology and Biosciences, University of Milano-Bicocca, Milan, Italy
2   Harvard Medical School and Division of Gastroenterology, Boston Children's Hospital, Boston, MA, USA
3   Department of Physics, University of Milano-Bicocca, Milan, Italy
4   Humanitas Clinical and Research Center, Rozzano (MI), Italy
5   Department of Microbiology and Immunobiology, Harvard Medical School, Boston, MA, USA
6   The Ragon Institute of MGH, MIT, and Harvard, Cambridge, MA, USA
   *Corresponding author. Tel: +1 6179196123; E-mail: ivan.zanoni@childrens.harvard.edu
   **Corresponding author. Tel: +39 0264483553; Fax: +39 0264483552; E-mail: francesca.granucci@unimib.it
   †These authors contributed equally to this work

NK-cell activation is dictated by the interaction with accessory cells producing activating mediators in both mice and humans. Dendritic cells (DCs) were the first described accessory cells capable of promoting IFN-γ production and cytotoxic activity by NK cells (Fernandez et al, 1999). Among the various DC-derived cytokines required for optimal NK-cell activation, IL-2 and IL-18 have been amply used in anti-tumor preclinical and clinical approaches (Skrombolas & Frelinger, 2014) (Fabbi et al, 2015). Interestingly, the therapeutic efficacy of these cytokines strongly correlates with their capacity to stimulate NK cells (Colucci et al, 2002).

Although we now have substantial understanding of the mechanisms underlying NK-cell activation, NK-cell trafficking between secondary lymphoid organs and inflamed or tumoral tissues and the dynamics of NK-cell interactions with accessory cells have been largely overlooked.

For instance, it is known that in resting conditions, NK cells reside in tissues and lymph nodes. NK cells home to lymph nodes via high endothelial venules (HEVs) and, presumably, also via the afferent lymphatics. The latter speculation is based on the observation that NK cells are present in the afferent lymph drained from healthy tissues (Carrega et al, 2014). In inflamed conditions, NK cells can be directly activated and recruited by DCs at the draining lymph nodes (Martin-Fontecha et al, 2004; Zanoni et al, 2005; Lucas et al, 2007). Whether lymph node-resident or newly recruited NK cells acquire anti-tumor effector functions following interaction with activated DCs is still unclear. It remains also to be determined whether anti-tumor effector functions can be exerted by tissue- or tumor-resident NK cells activated by accessory cells outside the secondary lymphoid organs. Moreover, while DCs and NK cells have been observed to form immune synapses (IS) enriched in surface and soluble molecules required for the activation of NK cells (Borg et al, 2004; Barreira da Silva et al, 2011) in vitro, in vivo studies have not confirmed the existence of long-lived interactions between these two cell types (Beuneu et al, 2009).

In the present work, we provide evidence that during an LPS-mediated inflammatory response, only NK cells resident in the LPS-draining lymph node at the moment of LPS administration are efficiently and transiently activated. Also, activated NK cells exert anti-tumor functions by leaving the lymph node and reaching the tumor. The activation of lymph node-resident NK cells is transient unless the inflammatory stimulus persists. When repeated injections of LPS are made to protract inflammation, NK cells functions are boosted, leading to persistent anti-tumor activity. Finally, we show that in order to induce NK-cell activation, prolonged interactions are established between NK cells and DCs at the lymph node draining the inflammatory stimulus. The prolonged contacts between DCs and NK cells allow DC-derived IL-18, produced at very low levels, to efficiently activate DC-interacting NK cells.

## Results

### DC-mediated activation of anti-tumor NK-cell functions occurs in the lymph node

The importance of the lymph node as a site of activation of anti-tumor NK-cell effector functions was evaluated in vivo. Mice were transplanted in the deep derma with the colon carcinoma CT26 cell line and 1 day later injected s.c. with LPS in the area draining the tumor site. Following a single LPS administration, a significant reduction in tumor growth was observable 5 days after tumor implant (Fig 1A). The tumor growth reduction was due to the effector functions of NK cells activated by DCs. Mice that were depleted of either CD11c+ or NK cells lost the protective effect of LPS administration (Fig 1B), and the same reduction of tumor growth was observed in SCID mice lacking T, NKT, and B cells (Fig 1C). In agreement with a transient capacity of LPS to activate NK cells, the effect of a single injection of LPS on tumor growth was short-term and was already lost 8 days after tumor injection (Fig 1A). Administering LPS every 4 days induced a persistent decrease in tumor expansion (Fig 1D). Following peripheral LPS administration, increased numbers of NK cells were observed inside the tumor. Also in this case, the increase in tumor-associated NK cell was dependent on DCs (Fig 1E). Since IFN-γ has been shown to interfere with tumor angiogenesis, we measured the level of tumor vascularization in animals treated or not with LPS. LPS-treated animals showed not only a reduced tumor volume but also a decreased perfusion of tumor masses as shown by immunohistochemistry (Fig 2). To investigate whether the NK cells activated in the draining lymph nodes were, in fact, responsible for the anti-tumor effect, the egress of cells from lymph nodes was reduced by treating mice with high doses of FTY720 (Mayer et al, 2004) before LPS treatment. FTY720 is known to reduce NK cell egress from secondary lymphoid organs by binding and downregulating the S1P receptor family, especially S1P1 (Matloubian et al, 2004), without altering NK cell effector functions, including cytokine production and cytotoxic activity (Jenne et al, 2009). Consistent with the literature, we confirmed that FTY720 administration did not interfere with NK-cell activation, while it reduced activated NK-cell egress from the draining lymph node (Fig 3A and B). Mice were therefore treated with FTY720 and/or LPS as depicted in Fig 3C, and tumor volume evaluated at Day 5. No anti-tumor effect (Fig 3C) and no increase in IFN-γ+ NK cells within the tumor (Fig 3D) were observed after FTY720 treatment. These data support a model in which DCs activate NK cells within the lymph node in response to LPS stimulation leading to subsequent NK-cell egress and manifestation of anti-tumor function at the tumor site.

### LPS does not induce NK-cell recruitment early after administration and favors activation of NK cells already resident in the lymph node

We then investigated whether the NK cells activated in the lymph nodes and responsible for the anti-tumor effect following LPS administration were already resident in the lymph node or newly recruited. For this purpose, mice were first transplanted with tumor cells and then treated with LPS 1 day after tumor cell injection. The entry of NK cells in the lymph nodes through HEVs was blocked by systemically treating the mice with the anti-CD62L antibody 24 h before LPS administration. We observed that total lymph node cellularity decreased (Appendix Fig S1A) indicative of the fact that lymphoid cells continuously recirculate. Since absolute NK-cell numbers also strongly diminished (Fig 4A), these data demonstrate that also NK cells recirculate like other leukocytes. The percentage of NK-cell numbers in the lymph node, nevertheless, increased by 5 times (Appendix Fig S1B), in both draining

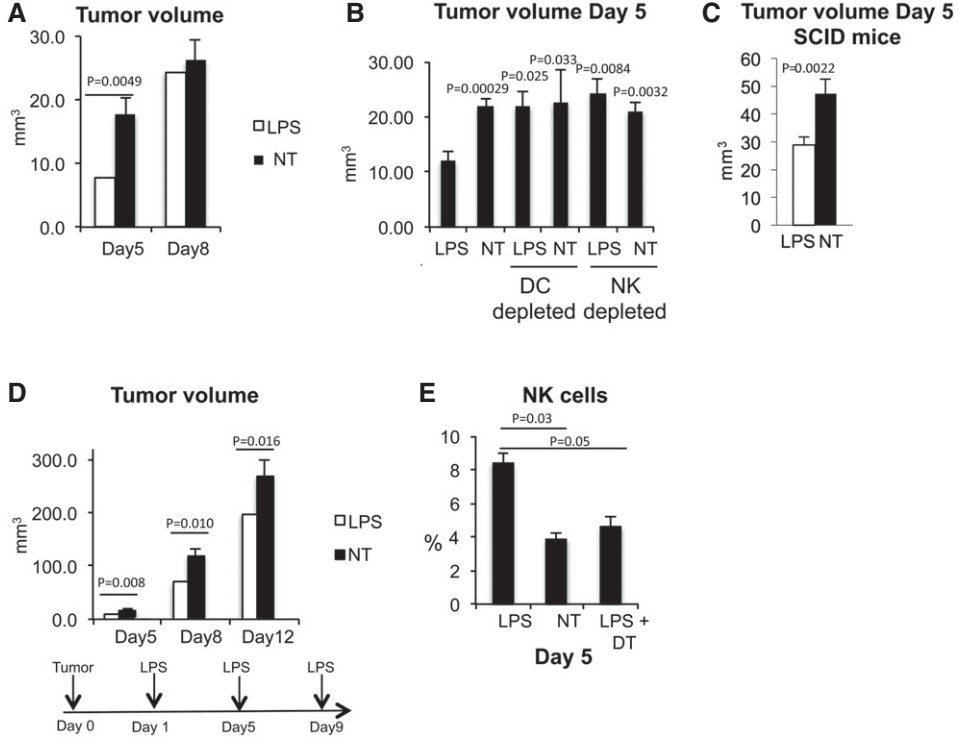

**Figure 1. Anti-tumor effector functions of NK cells after LPS administration.**

A  Tumor volume at the indicated time points after CT26 cell transplant in mice treated or not with LPS, *n* (number of animals per group) = 6.
B  Tumor volume in mice depleted of DCs or NK cells before tumor cell transplant and LPS administration, *n* (number of animals per group) = 5.
C  Tumor volume in SCID mice treated or not with LPS, *n* (number of animals per group) = 5.
D  Tumor volume at the indicated time point after repetitive LPS injections as depicted in the bottom scheme, *n* (number of animals per group) = 6.
E  Percent of NK cells within the tumor in mice treated or not with LPS and depleted or not of DCs, *n* (number of animals per group) ≥ 4.

Data information: NT, control; DT, diphtheria toxin-treated animals to deplete DCs; LPS, mice treated with LPS. Error bars depict SEM. Statistical significance was determined with a two-tailed *t*-test. NT, untreated.

and non-draining lymph nodes. These data suggest that the rate of recirculation at the steady state of other cell types, such as T cells, is higher compared to NK cells and potentially exclude a major role for PSGL-1/P-selectin-mediated LN recruitment of NK cells in our mouse model. When we analyzed the absolute numbers of IFN-$\gamma^+$ NK cells in the draining lymph node, we observed that inhibiting HEV-dependent cell recruitment to lymph node did not hamper NK-cell activation (Fig 4B). In particular, although the absolute numbers of IFN-$\gamma^+$ NK cells at the steady state were changed upon antibody treatment due to the reduced numbers of NK cells present in the lymph node, the numbers of newly activated NK cells after LPS administration (IFN-$\gamma^+$ NK-cell numbers in treated minus the numbers of IFN-$\gamma^+$ NK cells in untreated animals) were only minimally affected in mice treated with anti-CD62L. This suggested that blood-borne NK cells contributed only minimally to the pool of activated NK cells. The recruitment, over the regular homeostatic turnover, of NK cells from blood induced by LPS was observable after at least 12 hours (Fig 4A) and occurred after the egress of early-activated NK cells.

To further demonstrate that lymph node-resident NK cells are the ones preferentially activated at the time of LPS administration, CFSE-labeled NK cells were adoptively transferred at the same time

in which the mice received LPS. The presence of labeled NK cells and their activation were then evaluated 5 hours after LPS administration in the draining and the contralateral lymph nodes. The prediction was that if NK cells were actively recruited by LPS at the inflamed lymph node for their activation, a larger number of CFSE$^+$ cells should have been found at the draining lymph node compared to the contralateral one. We observed that labeled NK cells injected at the time of LPS administration reached the draining and the contralateral lymph nodes with equal efficiency, suggesting homeostatic turnover rather than active recruitment (Fig 4C). This excludes that LPS favors NK-cell recruitment early after administration and suggests that NK cells do not need to be newly recruited from the pool of circulating cells to be activated (Fig 4C). Also, fractions of IFN-$\gamma^+$ NK cells observed in the CFSE-positive and CFSE-negative populations were comparable in the LPS-draining lymph node (Fig 4D). This again confirmed the prediction that NK cells are not newly recruited to the inflamed lymph node to be activated. These experiments support the scenario that NK cells resident in lymph nodes at the time of stimulus administration are the cells that preferentially undergo activation.

To investigate whether the activation of lymph node-resident NK cells was sufficient to reduce tumor growth, we measured the tumor mass at Day 5 in anti-CD62L-treated animals. Figure 4E shows that

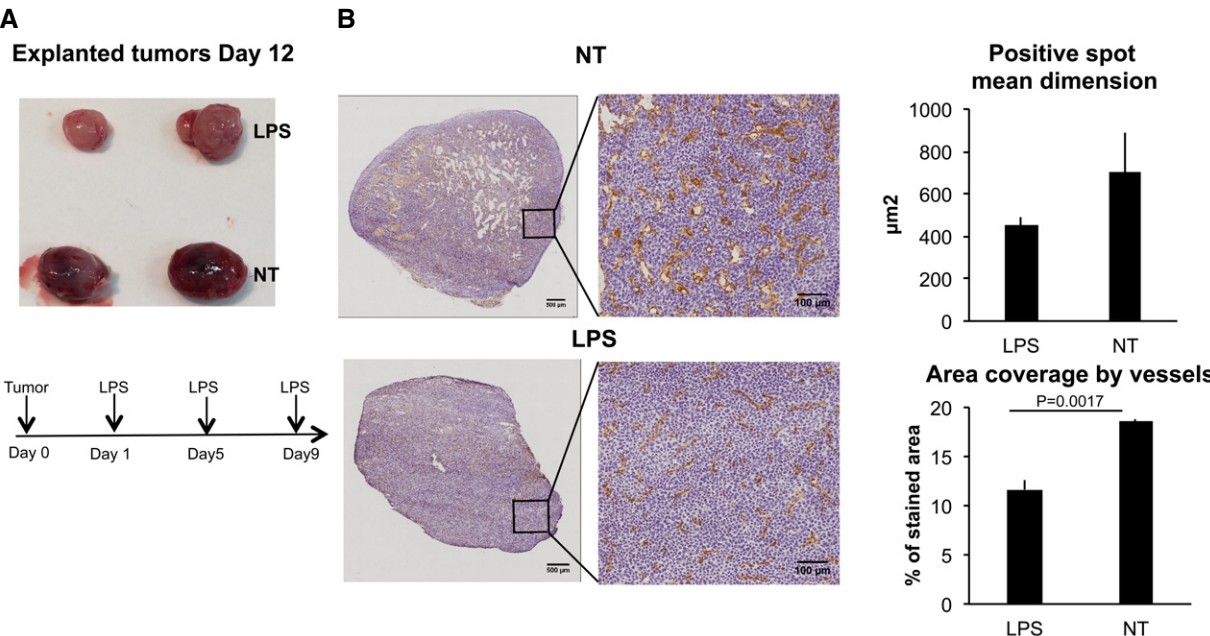

**Figure 2.  Effect of LPS on tumor vascularization.**

A  Explanted tumors at Day 12 after repetitive injections of LPS as depicted in the scheme at the bottom of the panel.
B  Detection of blood vessels by immunohistochemistry using anti-CD31 antibody (brown staining) in frozen sections of tumors from mice treated or not with LPS as in the scheme in (A). Nuclei are stained with hematoxylin. Two representative sections are shown, magnification 20×. Left-panel scale bars, 500 μm; right-panel scale bars, 100 μm. Quantification of vessel mean dimensions and area coverage by vessels is also shown. Three sections from three tumors were analyzed. Error bars depict SEM. Statistical significance was determined with a two-tailed *t*-test. NT, untreated.

blocking NK-cell recruitment to the lymph node does not hamper NK cell-dependent control of tumor growth. Accordingly, an increase in IFN-γ$^+$ NK cells could be observed inside the tumor (Fig 4F). These data suggest that lymph node-resident NK cells that are activated in response to LPS injection are sufficient to control peripheral tumor growth.

Taken together, these results indicate that following LPS administration, lymph node-resident NK cells are unique for their capacity to become effector cells by interacting with CD11c$^+$ cells. DC-activated lymph node-resident NK cells egress the lymph node, reach the tumor, and are preferential effectors in the control of tumor growth.

### During inflammation, DCs make prolonged contacts with NK cells in lymph nodes

Based on the above results, the lymph node represents the principal site of NK-cell activation. Lymph nodes are secondary lymphoid organs where cell–cell contacts are favored due to the high cell density. It has been repeatedly proposed that analogous to what happens for T cells, a productive DC–NK-cell cross talk requires cell–cell interaction and the formation of immune synapses where NK cell-activating molecules are concentrated and secreted, likely in a polarized way (Borg *et al*, 2004; Barreira da Silva *et al*, 2011). This requirement has been, then, questioned since no stable DC–NK-cell contacts have been identified *in vivo* when polyI:C or LPS was used to activate DCs (Beuneu *et al*, 2009). Due to the observation that lymph nodes are preferential

sites for NK-cell activation by accessory cells, we have reconsidered the possibility that long-lasting interactions between DCs and NK cells could occur in these secondary organs specialized for lymphocyte activation.

We decided, therefore, to re-analyze the capacity of these two cell types to interact in lymph nodes *ex vivo*. We reasoned that potential long-lived interactions between DCs and NK cells might have been overlooked simply because of their low frequency, as suggested by the low percentage, around 10–12%, of NK cells undergoing full activation in LPS-draining lymph nodes (Figs 3 and 4) (Zanoni *et al*, 2013).

The distribution and the motility of NK cells were assessed by means of two-photon microscopy in explanted lymph nodes as described previously (Caccia *et al*, 2008). Mice were adoptively transferred with fluorescently labeled cells, which populated the popliteal and brachial lymph nodes about 24 h after injection and persisted for at least 72 h. Imaging was performed in explanted lymph nodes (Caccia *et al*, 2008), and cell tracking was performed by means of the 3D tracking routine the Volocity software (Rush *et al*, 2009). Cellular dynamics can be described by several parameters, and for many of them, we noticed high consistency between our data and published reports, thus validating our approach. For instance, in agreement with previous studies showing the location of endogenous NK cells (Beuneu *et al*, 2009), we found that also transferred exogenous NK cells distributed mainly at the border of the T-cell area in Appendix Fig S2. We also confirmed that non-activated NK cells were highly motile in steady state conditions. Indeed, mean three-dimensional (3D) velocities of transferred NK cells, as measured in our experiments (Appendix Fig S3A),

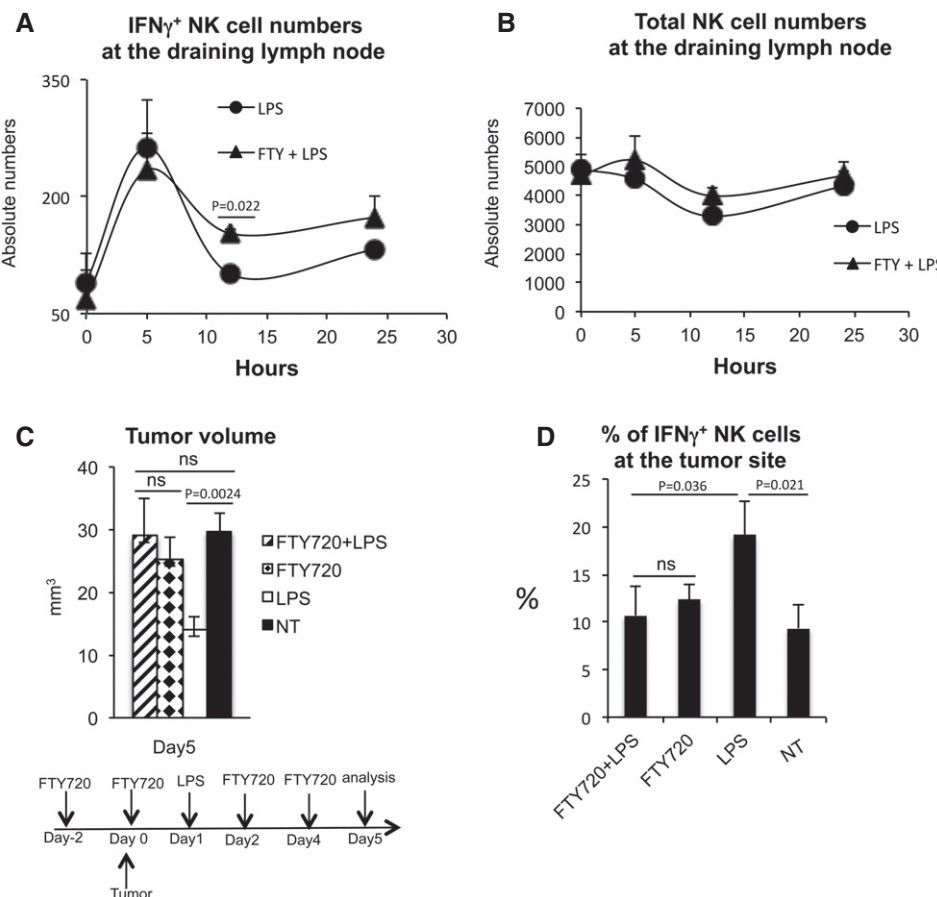

**Figure 3.  Anti-tumor effector NK cells are generated in the lymph node.**

A, B   Absolute numbers of IFN-γ⁺ and total NK cells in draining lymph nodes at the indicated time points before and after LPS administration in mice treated or not
        with FTY720 (25 μg/mouse) to reduce the ingress of lymphocytes in the efferent lymphatics.
C       Tumor volume in mice treated or not with LPS and/or FTY720 according to the scheme.
D       Percent of IFN-γ⁺ NK cells within the tumor in mice treated or not with LPS and with FTY720.

Data information: *n* (number of animals per group) = 4. Error bars depict SEM. FTY: FTY720. Statistical significance was determined with a two-tailed *t*-test. NT,
untreated.

overlapped with data available for endogenous NK cells (Beuneu
*et al*, 2009). Finally, as a further validation of our *ex vivo* experi-
mental setup on explanted lymph nodes (Villa *et al*, 2010), we
analyzed the motility of T cells. We found that their average instan-
taneous speed was $10 \pm 2$ μm/min (Appendix Fig S3B), in agree-
ment with available data (Miller *et al*, 2002). Given the good
correlation of our results with the literature, we concluded that the
*ex vivo* setup with adoptively transferred NK cells could be appropri-
ately used to define the behavior of NK cells and DCs in resting and
inflammatory conditions.

Intact brachial lymph nodes explanted from CD11c.GFP mice
(Jung *et al*, 2002), earlier on adoptively transferred with CMTPX-
labeled NK cells and treated or not with LPS, were imaged. Since
the kinetic of IFN-γ production showed a maximal peak of activa-
tion at 5 hours after LPS administration (Zanoni *et al*, 2013), we
started imaging lymph nodes between 2 and 4 hours after LPS
injection. Three different parameters, namely the confinement ratio
of NK-cell trajectory (defined as the ratio of the distance run by an

NK cell to the trace length), the instant velocity, and the NK-cell–DC
distance, were taken into account to define the dynamics of NK cells
and their possible interactions with DCs. These parameters and the
experimental conditions are extensively explained in the Appendix
(section "Analysis of DC and NK cell interactions"). In our hands,
each parameter, when taken alone, showed poor performance in
defining whether selected DCs and NK cells were interacting.
Conversely, by integrating information coming from each of these
three parameters, we could quantify with a user-independent algo-
rithm whether the selected couple of NK cells and DCs were inter-
acting. In particular, we set upper thresholds on the interacting NK-
cell speed ($\leq 60\%$ of the speed averaged over all the tracked cells in
a field of view; Appendix Fig S4) and the DC–NK-cell distance
($d \leq 25$ μm) and required that the confinement ratio was a non-
increasing function of the time (see Appendix for details;
Appendix Fig S5).

We define that a selected NK cell is interacting with a DC in the
i^th frame if the conditions on the instantaneous speed (Equation 4

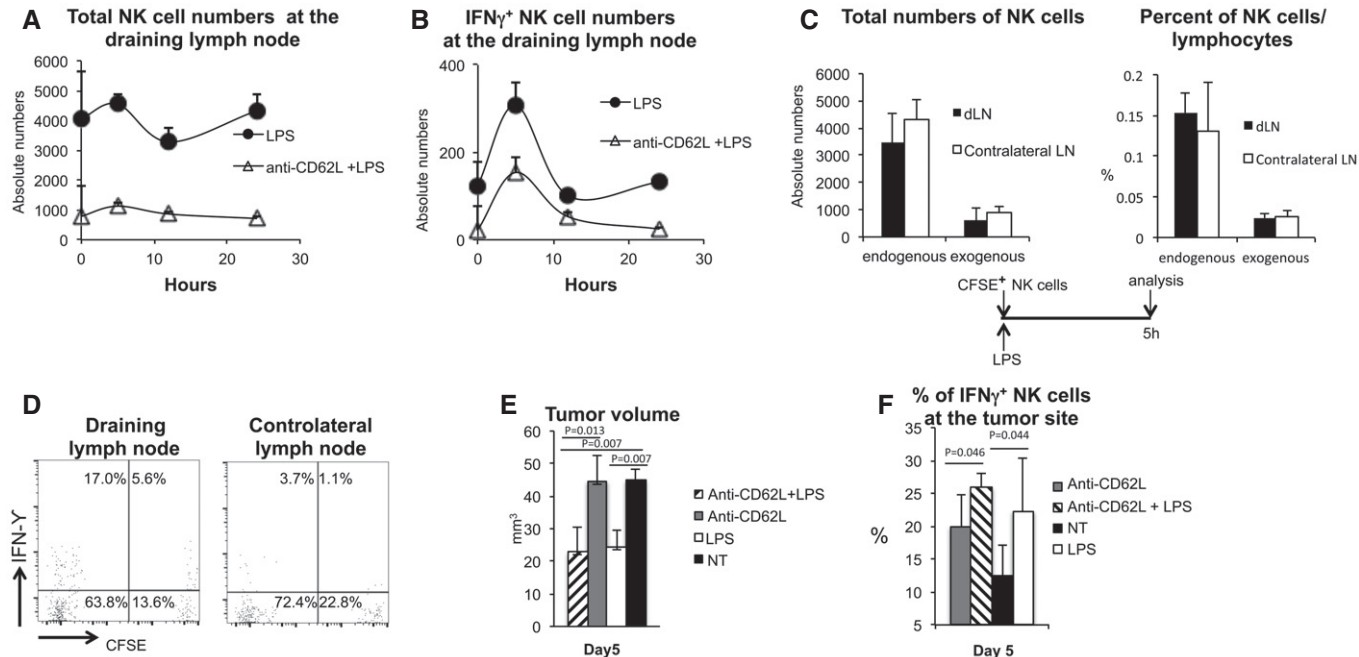

**Figure 4.  Blocking NK-cell recruitment at the draining lymph node does not interfere with the generation of effector NK cells.**

A   Absolute numbers of NK cells in one of the four draining lymph nodes in mice treated or not with the anti-CD62L antibody to inhibit NK-cell ingress in the lymph node from blood. Where indicated, mice were challenged with LPS.

B   Absolute numbers of IFN-γ⁺ NK cells in one of the four draining lymph nodes before and after LPS administration in mice in which NK-cell entry in the lymph node was blocked or not by anti-CD62L treatment.

C   (Left panel) Absolute numbers of endogenous and adoptively transferred (CFSE⁺) NK cells in the draining and contralateral lymph node 5 h after LPS administration; (right panel) percent of endogenous and adoptively transferred NK cells on total lymphocytes in the draining and contralateral lymph node 5 h after LPS administration.

D   Flow cytometry analysis of LPS-draining and contralateral lymph node from mice that received CFSE-labeled cells at the moment of LPS administration. Lymph node cells were stained with anti-CD3, anti-DX5, and anti-IFN-γ antibodies, and the analysis was performed on gated CD3⁻DX5^high cells. One representative of four independent experiments is shown.

E   Tumor volume in mice treated or not with LPS; where indicated, mice received anti-CD62L antibody 12 h before LPS treatment.

F   Percent of IFN-γ⁺ NK cells within the tumor in mice treated or not with LPS and with the anti-CD62L antibody.

Data information: (A, B, C, E, F) *n* (number of animals per group) = 4. Error bars depict SEM. Statistical significance was determined with a two-tailed *t*-test. NT, untreated.

in Appendix), the confinement ratio (Equation 8 in Appendix), and the NK–DC distance (Equations 1 and 2 in Appendix) are simultaneously met. Such frames can be highlighted in the plots of the confinement ratio, the normalized NK speed $v_{Norm}(t_i)$ (Equation 3 in Appendix), and the digitized NK–DC distance, $Dist_D(t_i)$ (Equations 1 and 2 in Appendix), by selecting those sets of contiguous frames for which the three following conditions are simultaneously met: The confinement ratio is a constant or decreasing function of time, $v_{Norm}(t_i) = 0$ and $Dist_D(t_i) = -0.15$. An example is given in Fig 5, where positively selected frames are marked by red striped boxes. The plots in Fig 5B and C report the interaction parameters over time of DCs labeled as DC1, DC2, and DC4 in the image (Fig 5A). Panel 5B corresponds to the absence of interactions between the NK cell (the red one in the image in Fig 5A) and the DC1, and panel 5C displays only short interactions, while panel 5D reports a long-lasting and a short-lived interaction. The duration of the interactions can then be estimated from these plots and used for statistical analysis. As discussed below, the definition of interacting events would be dramatically poorer if only one parameter at a time was used.

It is likely that this multiple conditioning of the data set slightly underestimates the number of the interactions but it largely reduces the number of false NK–DC interactions detected, and it allows us to measure the interaction times and perform their statistical analysis. Moreover, the algorithm offers the possibility to single out interactions on a quantitative and reproducible basis.

Using plots like the ones shown in Fig 5, we identified DC-interacting NK cells (Fig 6A). We found that around 45 ± 4% of the tracked NK cells interacted with DCs at the steady state. This percentage increased slightly after LPS administration, reaching 52 ± 6%.

We then focused on cells displaying contacts lasting more than 900 seconds (long interactions). At the steady state, we observed that most interacting NK cells displayed "touch and go" interactions (< 5 s) (Movies EV1 and EV2). On average, only 1% underwent long interactions with DCs. Diversely, in the presence of LPS, a relevant fraction of NK cells (12 ± 2% of interacting cells) experienced long interactions with DCs. In many cases, these interactions lasted more than 1/2 h (Fig 6B; see also Movies EV3–EV6). However, the mean 3D velocities of interacting cells were very similar, regardless

of the presence or absence of LPS (Fig 6C). This is presumably because cells showing long interacting time are too few to impact on the general mean velocity.

In summary, prolonged interactions between DCs and NK cells occur in the peripheral T-cell area of the lymph nodes in LPS-induced inflammatory conditions.

### IL-18 is the proximity-dependent signal required in DC–NK-cell interactions

The two-photon microscopy studies described above indicated that a productive DC–NK-cell interaction demands the formation of stable contacts. That requirement might be interpreted as dependence on a plasma membrane protein on DCs that must interact with an NK-cell receptor. Nevertheless, others and we observed that in the presence of LPS, only soluble—and not membrane-associated—DC-derived molecules are required for NK-cell activation. In particular, IL-12, IL-18, and type I IFNs in humans (Ferlazzo *et al*, 2004; Chijioke & Munz, 2013) and IL-2, IL-18, and type I IFNs in the mouse (Zanoni *et al*, 2013) are necessary and sufficient to drive DC-mediated NK-cell activation. Type I IFNs are in turn necessary to induce IL-15 production (Lucas *et al*, 2007).

The alternative explanation for the prolonged interaction observed between DCs and NK cells in inflammatory conditions in the lymph node may be that certain soluble factors are released in such low amounts to be effective only in the limited space surrounding the secretion source, where the local concentration is high enough. In this way, they would behave as proximity-dependent signals, despite their soluble nature.

We previously demonstrated that among the factors required for NK-cell activation, IL-12, IL-2, and type I IFNs are produced by DCs in sufficient amount to work on bystander cells (Zanoni *et al*, 2013).

A candidate cytokine possibly requiring synapse formation to be effective is IL-15. IL-15 is described to be required for NK-cell activation in the presence of LPS and transpresented in association with IL-15Rα. Nevertheless, others and we have shown that both in humans and in mice, IL-15 can act in an autocrine way on NK cells

expressing the IL-15Rα via *cis*-presentation (Zanoni *et al*, 2013; Mattiola *et al*, 2015), therefore excluding the necessity of DC–NK-cell contact.

Interestingly, it has been shown that IL-18 is secreted at the immune synapse between human DCs and NK cells (Gardella *et al*, 1999; Semino *et al*, 2005). Thus, it could well exhibit the properties of a contact-dependent, although soluble, factor. The precursor of IL-18 is almost ubiquitously expressed, while active IL-18 is generated in very limited amount only in specific cells, including DCs and macrophages (Dinarello *et al*, 2013).

*In vivo* and *in vitro*, IL-18 produced by DC is required for NK-cell activation. In our previous work (Zanoni *et al*, 2013), we observed that the activation of NK cells following LPS administration was severely compromised in chimeric mice in which only DCs were deprived of the ability to produce IL-18. Accordingly, no NK-cell activation could be observed *in vitro* in the presence of LPS when IL-18R-deficient NK cells or IL-18-deficient DCs were cocultured with WT DCs or WT NK cells, respectively.

We first investigated whether IL-18 is secreted at the contact site between DCs and NK cells also in the mouse system. These two cell types were cocultured in the presence of LPS, and the localization of IL-18 was analyzed by immunofluorescence after 2 h of incubation. As shown in Fig 7A, IL-18 redistributed at the contact sites between the two cell types in LPS-treated cocultures.

To test whether IL-18 was the contact-dependent signal required in DC-mediated NK-cell activation, we reasoned that naturally secreted IL-18, once dispersed in the culture medium, should be ineffective due to dilution. Therefore, we expected supernatants collected from LPS-stimulated DCs to be unable to elicit IFN-γ release by NK cells, unless rIL-18 was added. Indeed, that was proven to be the case (Fig 7B). Similarly, in a transwell culture system used to separate DCs and NK cells, IL-18 had to be added to the compartment containing NK cells in order to see their activation (Fig 7C).

These data establish that IL-18 is a proximity-dependent signal that requires a prolonged contact between DCs and NK cells to fully exert its activity.

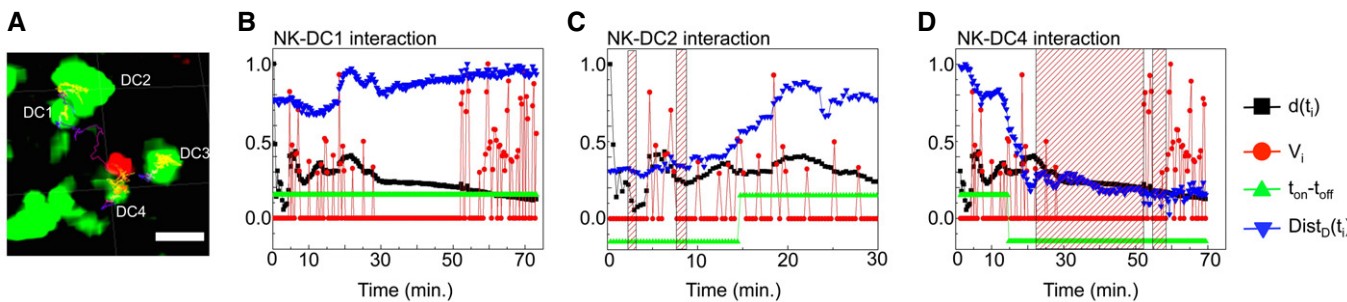

**Figure 5.  DC–NK-cell interactions in lymph nodes.**

A    Representative image of an NK cell (red) interacting with four DCs (green).

B–D  Plot of the confinement ratio (CR, black squares), the normalized NK speed (red circles), and the digitalized NK–DC distance, Dist$_D$(t$_i$) (green triangles), as a function of time for the NK cell reported in the image. Panels (B–D) correspond respectively to the interaction with the DC1, DC2, and DC4 displayed in (A). For reference, also the normalized distance Dist(t$_i$) is reported in the plots (blue down-triangles). The time periods in which all the three conditions discussed in the text are fulfilled are marked with red striped rectangles in the plots. The green up-triangles indicate the parameter $T_{on}T_{off}$ defined as $T_{on}T_{off} = -1$ if Dist(t$_i$) < d = 25 μm and $T_{on}T_{off} = -1$ if Dist(t$_i$) ≥ d and indicate the putative interactions according to a more simplified algorithm based on the DC–NK-cell distance alone.

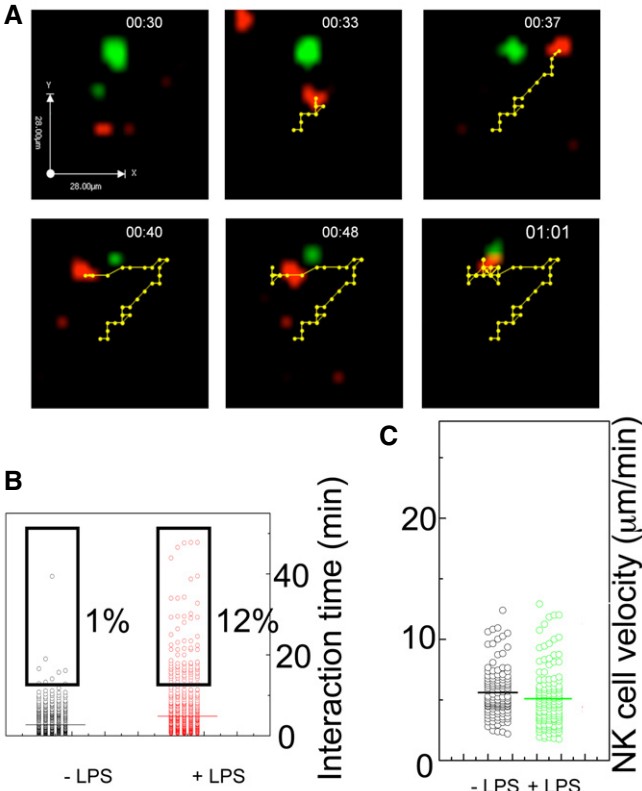

**Figure 6.** DCs and NK cells establish prolonged interactions in lymph nodes upon LPS injection.

A   NK and DC trajectories as they are tracked by Volocity software on the 4D volumes collected in lymph nodes on the TPE microscope. The sequence of images shows the interaction between an NK cell (red) and a DC (green). In the top panels, the NK cell moves according to a directional random motion, while in the lower left and middle panels, the NK cell seems to recognize the DC and to go back. In the right lower panel, the stable contact DC–NK cell is clearly visible.

B   Time duration of the DC–NK contacts in steady state (−LPS) and inflammatory (+LPS) conditions. Bars indicate the mean of the distribution. The percent of cells showing long interaction times (≥ 15 min) is shown.

C   NK-cell 3D velocities. 3D velocities of NK cells taking contacts with DCs before (black) and 5 h (green) after LPS treatment.

## Discussion

Understanding NK-cell distribution, site of activation, and trafficking is necessary to define immunization protocols aimed at reactivating protecting and persisting host anti-tumor functions. In the present work, we show that after the s.c. administration of LPS in the area draining a tumor site, NK cells activate their anti-tumor functions. Draining lymph node-resident NK cells, and not cells newly recruited from blood or non-lymphoid tissue, are activated by DCs, leave the lymph node, reach the tumor, and decrease tumor growth. The effect is only temporary, and continuous LPS administrations are necessary to maintain NK-cell anti-tumor effector functions. The DC-mediated activation of lymph node-resident NK cells correlated with the formation of relatively long-lasting interactions between the two cell types. This allows IL-18, a DC-derived cytokine required for NK-cell activation and produced in extremely low amount, to

efficiently act in a paracrine way at the interface between the two interacting cells.

NK cells mediate tumor suppression exploiting different mechanisms, including direct killing of tumor cells and IFN-γ production. IFN-γ is, in turn, required to activate type I immunity (Lee et al, 2015) and to interfere with tumor neovascularization (Curnis et al, 2005; Deng et al, 2014). CT26 cells are relatively insensitive to IFN-γ-induced type I immunity and cytotoxicity since IFN-γ downregulates a CD8[+] cell epitope favoring tumor escape (Beatty & Paterson, 2000); therefore, the anti-tumor effect mediated by NK cells is presumably not due to the activation of adaptive immunity. Accordingly, we observed that the LPS-mediated anti-tumor effect was present also in SCID mice lacking T and B cells. The primary IFN-γ-dependent anti-tumor effect exerted by NK cells in this model is presumably the anti-angiogenic function. We observed that the activation of NK cells significantly affected tumor vascularization with LPS-treated animals showing a reduced amount of vessels perfusing the tumor with a more regular distribution. The effect on tumor vascularization could be directly due to an anti-angiogenic effect of IFN-γ. An intriguing alternative is that IFN-γ release at the tumor site by activated NK cells could exert an inhibitory effect on tumor-associated neutrophil (TAN) recruitment. Evidence in the literature is present showing an inhibitory effect of NK cell-derived IFN-γ on neutrophil recruitment at the inflammation site in autoimmune and infection-mediated inflammatory conditions (Feng et al, 2006; Figueiredo et al, 2007; Wu et al, 2007). Further experiments will be performed in the future to confirm the existence of one or both scenarios.

In our study, the lymph node draining the inflammatory stimulus is the site of activation of NK cells. Blocking blood-borne NK-cell entry in the lymph node does not affect NK-cell activation and anti-tumor NK-cell functions, indicating that lymph node-resident NK cells at the moment of LPS activation are the first cells to be activated. After activation, NK cells leave the lymph node to reach the tumor site. Reduction of activated NK-cell exit from the inflamed lymph node completely eliminates the NK-cell anti-tumor functions observed after LPS treatment, indicating that the lymph node is the exclusive site of NK-cell activation in these conditions. Based on studies describing the presence of CD56[bright] perforin[low] in human seroma, it has been hypothesized that NK cells can leave the tissues and reach secondary lymphoid organs via the lymph (Carrega et al, 2014). Although we cannot formally rule out that some NK cells can be recruited from skin and arrive to the draining lymph node via the afferent lymphatics, we tend to exclude that those cells are the ones predominantly contributing to the observed anti-tumor NK-cell functions. LPS is quickly cleared from the skin by immediate drainage to lymph nodes as well as local inactivation by de-acylation (Lu & Munford, 2011).

In the lymph node, NK cells distribute in the T-cell area where DCs are present in high concentration forming an intricate net of cells (Beuneu et al, 2009). The high cell density is a strategic organization to favor cell–cell interactions. In this context, T cells, probing one DC after another, increase the probability to find the DCs equipped for their activation and establish long-lasting interaction with these cells. Since the lymph node represents the principal site of NK-cell activation, we predicted that analogous to what happens for T cells, also NK cells needed to find the DCs correctly prepared for their activation and to establish long-lasting interactions with them.

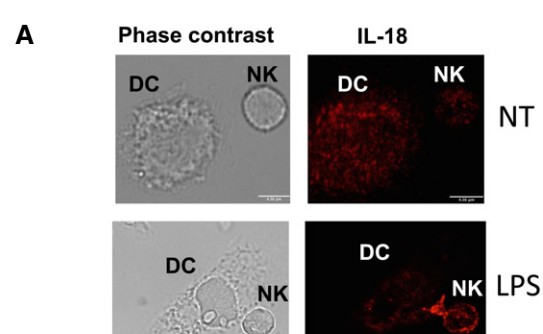

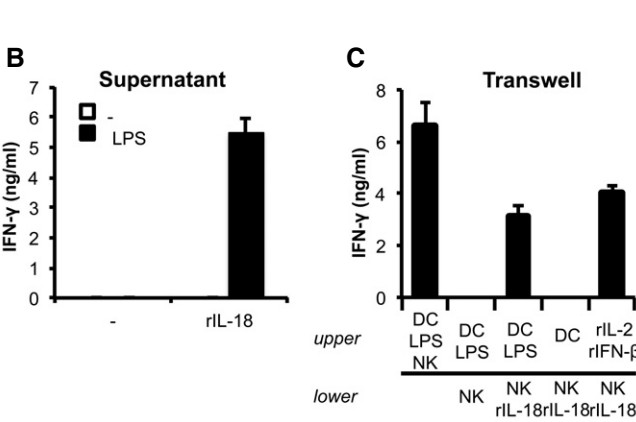

**Figure 7.   IL-18 is the contact-dependent signal required in DC–NK-cell interactions.**

A   Differential interference contrast images (left panels) and confocal microscopy analysis of IL-18 (right panels) in a DC/NK conjugate after 2 h of coculture in the presence of LPS and in unstimulated cells (NT). Representative images from 3 experiments are shown, magnification 63×. Scale bars, 4.36 μm.

B   IFN-γ release by NK cells cultured in the presence of supernatants recovered from untreated DCs (untreated) or DCs treated O/N with LPS (LPS). Where indicated, IL-18 was added to the cultures.

C   Cell contact-dependent activation of NK cells by DCs depends on IL-18. Unstimulated or LPS-activated DCs and NK cells were cocultured in the same wells (DC + NK + LPS) or separated by a porous membrane [(DC + LPS)/NK]. Where indicated, IL-18 was added to the transwells [(DCs + LPS)/(NK + IL-18)]. NK cells alone were also cultured in the presence of the three selected cytokines: IL-2 and IFN-β in the upper chamber and IL-18 in the lower chamber. IFN-γ in the supernatant was then measured by ELISA after 8 h of coculture.

Data information: (B, C) *n* (number of independent experiments) = 3. Error bars depict SEM.

Different from a previous work (Beuneu *et al*, 2009), we found that following LPS administration, NK cells stabilize their contacts with DCs, most likely to allow enough time to receive the activating stimuli. Indeed, the analysis of trajectories revealed that NK cells show a decrease in their motility in inflammatory conditions when compared to the steady state condition. Moreover, NK cells interacting with DCs transition from a "touch and go" regime at the steady state, in which short and numerous contacts with DCs are formed, to longer, long-lived contacts (Fig 5). We were able to reveal DC–NK-cell long interactions, overlooked in previous reports (Beuneu *et al*, 2009), thanks to the definition of a new method of measuring the NK–DC interaction times, possibly independent of the user direct inspection of the microscope movies and based on the computing and combination of a number of dynamic parameters. This choice was required for the highly dynamic behavior of NK cells displaying widely varying interaction times. We focused here on three parameters, namely the NK–DC distance, the instantaneous NK-cell velocity, and the confinement ratio. The rationale in the choice of the indicated parameters was that cell–cell interaction is mediated by the emission of chemical signals that imply NK cells lie in close proximity to a DC for a finite time stretch and display velocity slightly lower than the average. The requirement on the reduction of the confinement ratio further ensures that the interaction, though highly dynamic in character, lasts for a finite stretch of time.

The algorithm devised here enforces a strict series of constraints on the trajectory that avoid us to take as a real interaction a situation in which the NK–DC distance alone is low but the speed and the confinement ratio constrains are not satisfied. In our opinion, these situations correspond to instances in which the cells are not interacting but probably only passing by, as shown in the examples reported in the Appendix (Appendix Figs S6–S11). On the other hand, we have refined the computation of the interaction duration by specifically allowing that the fulfillment of the three criteria can fail for a single image and for a single parameter.

At first look, our data may seem to conflict with previous results by Beuneu *et al* (2009), who reported no or very few interactions lasting more than 5 min (300 s). However, careful inspection of their work indicates that the percentage of NK–DC interactions lasting more than 300 s (the maximum duration reported) at 2–4 h was about 12%. Although with some uncertainty, this estimate agrees with our results. Our analysis extends and refines previous analytical strategies, highlighting a small but finite fraction of long-lasting interactions positively correlating with cell biology data.

These data are in agreement with previous *in vitro* studies showing that NK cells and DCs form stable conjugates necessary for NK-cell activation, with the corresponding immunologic synapse enriched in adhesion molecules and inflammatory cytokines (Barreira da Silva & Munz, 2011).

Interestingly, it has been repeatedly shown that productive DC–NK-cell interactions require both soluble and contact-dependent signals (Fernandez *et al*, 1999; Newman & Riley, 2007). However, others and we demonstrated that soluble cytokines at physiological concentrations are able to fully activate NK cells in terms of IFN-γ production. This seeming discrepancy might be explained by the peculiar biology of IL-18. When the amount of released cytokine is low, a biologically active concentration might be achieved only in close proximity to the secretion source. However, upon dispersion, the concentration would drop to levels yielding a negligible activity. In this way, even soluble signals would behave like contact-dependent signals.

In agreement with the observed concentration of IL-18 at the cell contact sites, supernatants from LPS-stimulated DCs are not able to activate NK cells, unless supplemented with an excess of rIL-18, confirming that the confined secretion of IL-18 at physiological doses imposes a requirement of cell–cell contact in DC-mediated NK-cell activation (Fig 7).

Obviously, the IL-18-driven requirement for cell contact is in addition to adhesion molecules, such as integrins, building the architecture of the immune synapse (Sims & Dustin, 2002; Brilot

*et al*, 2007). Indeed, the formation of a stable physical interaction between accessory cells and lymphocytes is a precondition of activation, and DC–NK-cell interactions make no exception. Only when the structure of the immune synapse is in place, are cytokines polarized and released (Borg *et al*, 2004; Brilot *et al*, 2007). Therefore, disruptions of key adhesion molecules would likely result in loss of activation as well. However, this would merely be a consequence of the lack of structural (rather than functional) requirements. We did not investigate the molecular requirements at a structural level here; rather, our claim is that IL-18 needs cell contact at a functional level (Fig 7).

Overall, our findings shed new light on the complex events that lead to the development of effective NK cell-mediated responses. We found that DC–NK-cell interactions at the draining lymph node are essential to activate NK cells and that lymph node activated NK cells retain a unique capacity to mount effective anti-tumor responses. Blocking activated NK-cell egress from the lymph node completely abrogated the NK cell-protective effect against tumor growth. The development of anti-tumor therapies aimed at boosting innate immune lymphocyte functions should take into account these requirements in order to potentiate our capacity to fight tumor growth and spread.

# Materials and Methods

### Mice

All mice, housed under specific pathogen-free conditions, were female on a BALB/c background for at least 12 generations and were used at 7–12 weeks of age. WT and SCID animals were supplied by ENVIGO, Italy, and used at 7–12 weeks of age. DTR mice, expressing the diphtheria toxin receptor under the CD11c promoter, have been already described (Jung *et al*, 2002). Experiments were performed using protocols approved by the Institutional Animal Care and Use Committee of the University of Milano-Bicocca and by the Italian Ministry of Health. Mice were kept in a pathogen-free conventional animal house facility. The animal house is run by professional employees fully equipped with state-of-the-art instrumentation in order to maintain the standard of animal welfare at the maximum levels. All mice were housed in individual, ventilated cages (IVCs) with 12-h light/dark cycles with food and water *ad libitum*.

To prevent pain or distress, we avoided restraining animals for longer than 5 min at a time, for most of the procedures performed. For tumor cell injections, mice were anesthetized by i.p. injection with ketamine (80–100 mg/kg) and xylazine (10 mg/kg), premixed before application. Mice subjected to tumor injection were monitored on a daily basis for signs of discomfort, including hunched posture, ruffled fur, and lack of movement within the cage. The body condition score index (a qualitative assessment of an animal's overall appearance based on its weight, muscle mass, and bone prominence) was used to evaluate the welfare of the mice. Generally, mice did not present signs of distress given the short period of time between tumor injection and killing.

Each experiment depicted in the figures was repeated at least twice with at least four mice per group. Globally, around 600 BALB/c animals (WT or genetically modified) were used.

### Reagents

Hybridoma producing anti-L-selectin monoclonal antibody Mel-14 was grown and the antibody purified according to standard procedures. FTY720 was purchased from Sigma-Aldrich. TLR-grade Reform LPS from *E. coli* serotype R515 was purchased from Enzo Life Sciences. The antibodies used were as follows: anti-asialo GM1 (Functional Grade Purified, cat. no. 16-6507; eBioscience); PE anti-mouse CD49b (clone DX5, cat. no. 108908; Biolegend); APC anti-mouse IFN-γ (clone XMG1.2, cat. no. 505810; Biolegend); FITC anti-mouse CD3 (clone 17A2, cat. no. 100204; Biolegend); APC anti-mouse CD11c (clone N418, cat. no. 117310; Biolegend); APC/Cy7 anti-mouse CD3 (clone 17A2, cat. no. 100222; Biolegend); purified anti-mouse CD31 (clone MEC 13.3, cat. no. 102502; Biolegend); PE/Cy7 anti-mouse CD45.2 (clone 104, cat. no. 109830; Biolegend); and anti-IL-18 polyclonal antibody (cat. no. sc-7954-Y; Santa Cruz Biotechnology).

### Cells

All cells were cultured in IMDM-10 complete medium: IMDM, 10% heat-inactivated FBS (EuroClone), 2 mM L-glutamine, 100 U/ml penicillin, 100 μg/ml streptomycin, and 50 μM 2-mercaptoethanol (Sigma-Aldrich).

CT26 cell line (ATCC, CRL2638), a BALB/c mouse colon carcinoma, was obtained from Mario Colombo (Istituto Nazionale dei Tumori, Milan, Italy).

BM-derived DCs (BM-DCs) were generated by culturing BM precursors, flushed from femurs, in GM-CSF-supplemented medium for 8–10 days, as described (Inaba *et al*, 1992; Granucci *et al*, 2001).

NK cells and T cells were purified from RBC-lysed splenocytes by MACS-positive selection using CD49b (DX5), or CD4 and CD8 microbeads (Miltenyi Biotec), respectively. Purity was assessed by FACS and was routinely between 93 and 96% for NK cells, and at least 98.5% for T cells. Alternatively, NK cells were first enriched by MACS-negative selection for Ly-6G (1A8), CD19 (6D5), and CD3ε (145-2C11), using biotinylated Abs and streptavidin microbeads, then stained with anti-CD49b, and sorted with a FACSAria II (BD Biosciences). Purity was consistently greater than 99.5%. MACS-selected and FACS-sorted NK cells produced similar results in our experiments.

### *In vivo* experimental settings

For the analysis of anti-tumor effect of NK cells *in vivo*, mice were inoculated in the deep derma in the left flank with the minimal tumorigenic dose of CT26 tumor cells ($5 \times 10^4$) at Day 0 and with LPS (10 μg/mouse) at the tumor draining area at Day +1.

To deplete DCs, CD11c.DOG mice received only 1 injection of DT (300 ng/mouse) at Day −1. To deplete NK cells, mice received anti-asialo GM1 polyclonal antibodies (eBioscience, 30 μg/mouse i.v.) at Days −3, −1, and +1.

To measure IFN-γ$^+$ NK cells within the tumor, LPS has been re-injected at Day 5 and IFN-γ$^+$ NK cells enumerated 5–6 h after LPS injection.

In some experiments, mice were also treated with FTY720 (25 μg/mouse) or anti-CD62L (100 μg/mouse) antibody according to the scheme indicated in the figures.

Tumor volume (mm³) was calculated according to the following formula: (diameter1 × diameter2)$^2$/2.

### ELISA

Cell-free supernatants were collected from DC–NK-cell cocultures after 18 h, and they were analyzed using an IFN-γ DuoSet ELISA Kit (R&D Systems). We have previously demonstrated that IFN-γ is exclusively produced by NK cells in DC–NK-cell cocultures (Granucci *et al*, 2004).

### Immunohistochemistry

Explanted tumors were embedded in OCT freezing media (Bio-optica). Sections (5 μm) were cut on a Cryostat, adhered to Superfrost Plus slide (Thermo Scientific), fixed with acetone, and blocked with PBS containing 0.3% Triton X-100 (Sigma-Aldrich) and 10% FBS. Sections were then stained with purified anti-mouse CD31 antibody (1:200) in blocking buffer, 1 h at room temperature. Sections were washed with TBS buffer and incubated with Rat-on-Mouse HRP-Polymer kit (Biocare Medical). Dako EnVision System-HRP was used as chromogen and counterstain with Mayer's Hematoxylin (Bio-optica). After dehydration, stained slides were mounted with Eukitt and images acquired with a Vs120 dotSlide (Olympus). Image analysis was performed using FIJI-ImageJ (Schindelin *et al*, 2012, 2015). ROI color deconvolution was used to separate the DAB signal, and particle analysis was used to automatically recognize, count, and measure all the CD31-positive spots within the total tumor area.

### IL-18 detection in DC–NK-cell cocultures

Cells were mixed at an NK cell/DC ratio of 2:1 in 100 μl of RPMI1640 without serum and pelleted quickly at 228 rcf. The pellets were then allowed to conjugate for 2 h at 37°C in the presence or absence of LPS. After conjugation, cells were gently resuspended in RPMI1640 and centrifugated on coverslips for immunofluorescence analyses.

Cells were thus fixed in 4% paraformaldehyde for 10 min at 4°C and permeabilized with 0.1% Triton X-100 and 1% BSA for 10 min at room temperature. Cells were finally stained with the anti-IL-18 antibody in the presence of 5% serum, followed by the appropriate secondary reagent. All washes were performed in PBS supplemented with 1% BSA. After staining, slides were mounted with FluorSave™ (Calbiochem). Slides were visualized through a ×63 oil immersion lens with an inverted TCS SP5 (Leica). Serial optical sections (0.3 μm; 15–20 sections) were acquired.

### *In vivo* NK-cell activation and IFN-γ production

To activate NK cells, age-matched mice were injected s.c. with 0.5 μg LPS/gbw. Where indicated, mice received previously CT26 tumor cells or were treated with FTY (25 μg/mouse) or with anti-CD62L neutralizing antibody (100 μg/mouse).

Mice were euthanized at different time points, and single-cell suspension of lymph node, skin, or tumors was kept for 3 h, in the presence of brefeldin A (BFA, 10 μg/ml; Sigma-Aldrich).

Intracellular staining was performed using Cytofix/Cytoperm reagents (BD Biosciences) according to the manufacturer's instructions, with the following Abs: anti-CD49b (DX5 or HMα2), anti-CD3ε (145-2C11), anti-CD11c (HL3), and anti-IFN-γ (XMG1.2) (or its isotype control). Samples were acquired with a Gallios flow cytometer (Beckman Coulter).

Single-cell suspensions from tumors were obtained using the mouse Tumor Dissociation Kit from Miltenyi.

Absolute cell numbers were calculated using Flow Count Fluorospheres (Beckman Coulter) according to the manufacturer's recommendations.

### Two-photon microscopy

A direct optical microscope (BX51; Olympus) was used for intravital imaging of DC–NK-cell interactions.

The infrared laser source (Mai Tai HP+DeepSee, Spectra Physics, USA; with pulses of 120 fs full width at half maximum and 80 MHz repetition frequency) is coupled through the FV300 (Olympus, Japan) scanning head. All the measurements were acquired through a 20×, 0.95-NA, 2-mm-WD objective (XLUMPlan FI; Olympus, Japan). TPE allows limited photodamage of the samples, simultaneous excitation of multiple fluorescent probes, and deep penetration in thick tissues such as DLNs. The fluorescence signal is steered to a non-descanned unit and split into three channels (blue, green, and red channel) by two dichroic beam splitters (Caccia *et al*, 2008). Additional details on the setup and its optical characterization can be found in Caccia *et al* (2008).

For *ex vivo* experiments, the entire microscope is surrounded by a custom-made thermostatic cabinet in which the temperature is kept at 37°C (air thermostating by "The Cube"; Life Imaging Services, Basel, CH) and physiological conditions are guaranteed during the experiments by flowing 37°C buffer solutions saturated with a mixture of 95% O₂–5% CO₂.

Volocity (Perkin-Elmer Inc.) was used to analyze recorded movies. The extracted traces were then analyzed for the measure of the interaction time by means of a specifically designed MatLab (MathWorks Inc.) code.

### Statistical analysis

Means were compared by paired or unpaired *t*-tests. Data are expressed and plotted as means ± SEM values. Sample sizes for each experimental condition are provided in the figure legends. *P*-values are provided in the figures. All of the statistical analyses were performed in blind by a third person.

**Expanded View** for this article is available online.

### Acknowledgements

FG is supported by grants from the Associazione Italiana per la Ricerca sul Cancro (AIRC, IG14593), the Fondazione Cariplo (grant 2014-0655), ARISLA (grant DC-ALS), and Fondazione Regionale per la Ricerca Biomedica (FRRB). IZ is supported by grants from the Fondazione Cariplo (grants 2014-0859, 2013-0624), NIH grant 1R01AI121066-01A1, CCFA Senior Research Award (grant 412708), Harvard University Milton Fund, and HDDC P30 DK034854 grant.

## The paper explained

### Problem

Natural killer cells are innate immune cells that have anti-tumor, anti-viral, and anti-bacterial functions. They represent an early source of interferon-γ, a cytokine required for the development of anti-tumor type I inflammatory immunity. The outcome of anti-tumor vaccination protocols, therefore, strongly depends on the efficiency of activation of endogenous NK cells, and efforts are being made to manipulate them for immunotherapeutic approaches. Low levels of NK-cell activity usually correlate with a high risk to develop cancer, and high numbers of NK cells in the tumor microenvironment correlate with improved prognosis of tumor-affected patients. Dendritic cells (DCs) are required for NK-cell activation, and the mechanisms underlining this process have been well characterized. Yet there is little information concerning NK-cell trafficking between secondary lymphoid organ and inflamed or tumoral tissues and the site and dynamic of NK-cell interaction with accessory cells.

### Results

In this study, we provide evidence that following s.c. lipopolysaccharide administration, NK cells are not actively recruited at the draining lymph node at early time points, but continue their regular homeostatic turnover. Only NK cells resident in the draining lymph node at the time of stimulus administration, therefore, become activated and exert anti-tumor functions by leaving the lymph node and reaching the tumor. The activation of lymph node-resident NK cells is short-term unless the inflammatory stimulus persists. The activation of NK cells at draining lymph nodes correlates with the formation of stable and prolonged interactions with dendritic cells to allow IL-18, a cytokine produced at very low level by LPS-stimulated DCs, to efficiently stimulate DC-contacting NK cells.

### Impact

These findings provide important advances on the mechanisms of NK-cell activation *in vivo* in response to a major inflammatory stimulus. The understanding of the dynamics governing NK-cell responses will provide a framework for the generation of immunotherapeutic strategies aimed at boosting innate immune lymphocyte functions to fight tumor growth and spread.

## Author contribution

FG conceived the research and wrote the manuscript; IZ conceived and oversaw the project; FM conducted most of the experiments; TG, MCa, MCo, LS, and GC conducted the two-photon analyses; RS conducted the transwell experiments and helped with data interpretation; CC, MR, and MDG helped with NK-cell activation experiments; MS performed FIJI-ImageJ analysis of immunohistochemistry samples; and UHvA supervised the preliminary experiments showing NK-cell motility at the lymph node in resting conditions and edited the manuscript.

## Conflict of interest

The authors declare that they have no conflict of interest.

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
