## [Review Process File · EMBO Molecular Medicine]

Manuscript EMM-2015-06164

Prolonged contact with dendritic cells turns lymph node-resident NK cells into anti-tumor effectors

Francesca Mingozzi, Roberto Spreafico, Tatiana Gorletta, Clara Cigni, Marco Di Gioia, Michele Caccia, Laura Sironi, Maddalena Collini, Matias Soncini, Michela Rusconi, Ulrich H. von Andrian, Giuseppe Chirico, Ivan Zanoni and Francesca Granucci

Corresponding authors: Francesca Granucci, University of Milano-Bicocca and Ivan Zanoni, Boston Children's Hospital

Review timeline:

Submission date:	21 December 2015
Editorial Decision:	03 February 2016
Revision received:	05 May 2016
Editorial Decision:	01 June 2016
Revision received:	09 June 2016
Accepted:	20 June 2016

Editor: Roberto Buccione

Transaction Report:

1st Editorial Decision

03 February 2016

Thank you for the submission of your manuscript to EMBO Molecular Medicine. We have now heard back from the three Reviewers whom we asked to evaluate your manuscript.

Although the Reviewers agree on the potential interest of the manuscript, the issues raised are of a fundamental nature. I will not dwell into much detail, but I would like to highlight the main points.

You will see that Reviewer #1 laments the descriptive nature of the work presented and the lack of mechanistic insight. Reviewer #2, while more positive, does find insufficient experimental support for the main conclusions. Reviewer #3 finds that essential novelty is limited by an earlier study, but also that the potential differences (and advances) with respect to the previous study are not convincingly supported.

In conclusion, while publication of the paper cannot be considered at this stage, given the potential interest of your findings and after internal discussion, we have decided to give you the opportunity to address the criticisms. It will be important that you conclusively show that it is lymphnode NK and not circulating NK that mediate DC activation (reviewer #2 suggests a number of experimental approaches in this respect). You should also discuss, but preferably show, using another model or assay why you find that DC-NK interactions are not short-lived, at variance with previous findings. Indeed, your current discussion appears somewhat dismissive of earlier studies. Finally, a significant effort should be made to gain further mechanistic insight, especially with reference to the role of IL-18 as pointed out by both Reviewers #1 and 3.

We are thus prepared to consider a substantially revised submission, with the understanding that the Reviewers' concerns must be addressed with additional experimental data where appropriate and that acceptance of the manuscript will entail a second round of review. The overall aim is to significantly upgrade the relevance and conclusiveness of the dataset, which of course is of paramount importance for our title.

I understand that if you do not have the required data available at least in part, to address the above, this might entail a significant amount of time, additional work and experimentation and might be technically challenging, I would therefore understand if you chose to rather seek publication elsewhere at this stage. Should you do so, we would welcome a message to this effect.

***** Reviewer's comments *****

Referee #1 (Comments on Novelty/Model System):

Although the authors make a number of interesting observations they do not even attempt to understand the underlying molecular mechanisms. Therefore, this manuscript is not suited for publication in the Journal.

Referee #1 (Remarks):

The study analyzes the effect of LPS injections on the anti-tumor activity of NK cells. Tumor infiltration by lymph node-resident NK cells is enhanced by LPS injections leading to slower tumor growth. The authors observe an interaction of NK cells with dendritic cells. In addition, a stimulatory activity of IL-18 is documented.

The authors make a couple of potential interesting observations. The underlying molecular mechanisms should be worked out.

Major points:

1. The authors do not really show that IL-18 is DC derived as they state in the abstract.
2. The LPS is interesting but it has to be shown whether this is a direct or indirect effect. On which cells does LPS act. Which genes are stimulated and which mediators are released?
3. The authors treat mice with an anti-CD62L antibody to block the entry of NK cells to lymph nodes through HEVs. What else is the anti-CD62L antibody doing *in vivo*? This should be explained and discussed.
4. Have the plasma levels of cytokines been measured?
5. If IL-15 can act in an autocrine fashion, does this necessarily exclude an effect of this cytokine on NK cells? This should be explained and discussed.
6. What happens if the activity of IL-18 is blocked *in vivo*?
7. Have the authors done experiments with IL-18^{-/-} mice? These are available from Jackson Labs.

Minor points:

1. What is the molecular mechanism of FTY720? This should be explained and discussed.
2. Is the secretion of IL-18 stimulated via LPS on dendritic cells?
3. The attached movies need to be better explained.

Referee #2 (Remarks):

The manuscript addresses a fundamentally important question of where NK cells become activated *in vivo*, which is an important question for stimulating anti-tumor responses. The study proposes that lymph node NK cells and not circulating NK cells are responsible for anti-tumor responses following local administration of adjuvant, such as LPS. This

concept is novel and interesting. However, I feel there are some concerns that need to be addressed to corroborate this principle.

1) It is unclear whether all peripheral NK cells are CD62L positive and whether neutralization of CD62L entirely blocks the recruitment of peripheral NK cells into the lymph nodes. The authors should analyze CD62L KO mice and see whether NK cells are depleted in lymph nodes in steady state and after LPS injection.

3) Fig 3a. needs to be graphed with matched +/- FTY time points in order to evaluate whether administration of FTY influences NK cell function since it appears FTY does influence acquisition of IFN-g in contrast to the authors claims that it does not.

4) Fig 3,b I am confused as to whether egress/function are really affected here. What % of NK cells and IFNg+ NK cells are being presented? A % of total cells, or % of just NK cells? The % might be affected by a lack of leukocyte egress from the lymph node if FTY blocks. Therefore, it might be better to display absolute No's of NK and IFNg+ NK - is this possible? If not, please explain.

5) Fig 3c,d - difficult to ascertain whether the reduced tumor vol. and %IFN-g+ NK cells in tumors is due to reduced recruitment or activation.

6) Fig. 4. Cells recruited to the lymph node are blocked with anti-CD62L, but the authors do not find that the % IFNg+ NK cells is significantly increased even though there is definitely an increased trend. The authors need more samples to gain statistical significance before they can conclude there is an increase otherwise they cannot claim that resident NK cells are being activated, especially since tumor vol is decreased.

7) It's easy to say lymph node resident NK are preferentially activated if you've included an anti-CD62L blocking mAb to block cellular recruitment to the lymph node. Why can't NK cells that are freshly recruited to lymph nodes also subsequently become activated? In which case - what is the point of this experiment?

8) Fig 4c,d should come before c,d since this explains the above query.

9) Fig 4d. The experiment is proposed to show page 7 ³we evaluated whether NK cells could be directly activated in the skin and could reach the lymph node after activation..².

- I may be wrong, but I'm not sure there is any evaluation of whether NK cells can reach lymph nodes in this experiment because it is not clear whether the IFN-g+ NK cells present in the skin after LPS administration are resident or recruited or whether they have migrated to resident lymph nodes and that is why there is no difference in % IFN-g+ NK cells - can the authors please clarify what is happening here?

10) Figs 5 & 6 represent interesting new findings. How do the authors know that the CD11c+ cells they are analyzing are DC and not e.g. CD169+ macrophages? (Coombes et al. 2013. Cell reports) can this question be addressed to some degree?

Referee #3 (Comments on Novelty/Model System):

The techniques are great, but more statistical analysis required for the in vivo imaging, esp since the author have different conclusion versus previous study. Fairly novel in that very few studies of DC-NK interaction in vivo, but idea of NK-DC crosstalk in LN has been around for a decade. System is contrived an unlikely pertains the physiology of tumor immunity or infection. Not sure it is relevant to immune therapy.

Referee #3 (Remarks):

In the presented manuscript, the authors provide some evidence that NK cells become activated by DCs in the lymph node to exert their antitumoral functions after stimulation with lps. The results showing a direct in vivo interaction between NK cells and DCs in the lymph node are interesting and new. The authors have modified and improved their 2-photon method, resulting in convincing data that lps induces increased interactions between adoptively transferred NK cells and DCs. The authors show nicely that the lymph node is required for endogenous NK cell activation by lps stimulation. The kinetic experiments indicate that LN activated NK cells travel to the tumor to induce antitumor immunity. The authors argue that sustained interaction between DC and NK cell in the LN leads to IL-18 induced NK cell IFN production but these data are weak.

Overall, the data are convincing and the authors interpret the data well. The finding that lps induces NK cell IFN production in the LN is not novel. In fact, Martin-Fontecha and Sallusto published these results in seminal studies in *nature Immunology* 2004, but as discussed in the manuscript introduction, these results are likely due to recruitment of NK cells into the LN, and not due to LN resident NK cells. The authors posit that the novelty of their findings is that resident NK cells are activated by lps in their model system. This somewhat contradicts the Sallusto findings and could be due to differences in the model system. There is a wealth of literature discussing the role of NK cell - DC crosstalk. The authors results add to this concept by showing direct in vivo interactions between NK and DC in the LN. In my opinion this is the most novel aspect of the data, although the results contradict a previous study that showed that NK-DC interactions were short-lived. The authors discuss this difference. These two apparent contradictions to the literature highlight how context-dependent the authors' results are. One way to address this issue is to use another model system, although this could be beyond the scope of the study.

Specific comments

In the methods, lps was injected sc in some exps and iv in others. This could be better defined in the figures and figure legends.

Fig 1:

- The authors should not use "CTL" to abbreviate "control". It is used to abbreviate cytotoxic T lymphocytes.
- Are T cells already active in the cancer model used? The protective function of LPS could also be lost in the DC-depleted mice because of loss of T cell function

Fig 2:

- In b, please correct "untreated" into "untreated".
- Please provide a scale bar for the microscopic pictures
- How many tumors were assessed?
- What is the relevance of this figure?

Fig: 3

- interesting data
- need to discuss that effect of FTY on tumor growth could be due to blocking ag-specific T cell migration
- cannot conclude that NK cell antitumor activity is due to their IFN production in the tumor. It is possible that NK cells make IFN to polarize to Th1 in the LN, and the T cells are required for tumor rejection.

Fig 4:

- Anti-L-Selectin is supposed to inhibit the migration of NK cells into the lymph nodes. Here, the % of NK cells is increased. This is probably due to the antibody inhibiting the ingress of T cells, which increases the percentage of NK cells. The authors should show the data for total NK cells. Only this would show that the antibody actually inhibited the ingress of NK cells from blood into lymph nodes.

- In fact, the authors should show the effect of lps on total LN cellularity since previous studies show that LN size increased after adjuvant injection.

- Authors should show that anti-CD62L does NOT affect IFN γ NK cell in TUMOR to confirm that ingress in LN is not important

-

Fig 5

If I understand correctly, the authors show examples of 3 different types of DC-NK interactions after lps injection. Why is there heterogeneous interactions? Is this dependent on the timing of the lps?

Fig 6

-this doesn't look significant to me

-why did the authors only plot 2 parameters when 4 are used in fig 5?

-what timepoint did the author choose to represent the velocity? According to fig 5, the velocity can be drastically different (almost 5X difference in 70 min time frame)

Fig 7:

- The title of this figure is not supported by the data.

- The authors should show that LPS-stimulated DCs indeed secrete IL-18 in their hands. If they don't show that, they cannot say that these cells are insufficient to induce IFN release by NK cells. Maybe the release of IL-18 by DCs did just not work in their experiment.

- This experiment is insufficient to make the conclusion "IL-18 is the contact-dependent signal required in DC-NK cell interactions". So far, the authors have only shown that it is sufficient. The contact-dependence is in no way shown, b suggests both (NK cells can still be activated even though they don't come in direct contact with DCs, as long as DCs are stimulated with LPS). The authors should perform additional experiments.

- In fact, I would interpret the data to show that lps+DC induces a soluble factor that can activate NK cell IFN γ in the presence of rIL-18 given to the NK cell.

- The dependence on IL-18 is not shown. DCs deficient in IL-18 would be the definitive experiment here.

- Title should be changed or more experiments done.

Minor comments

- The legends in Figures 3c and 4c are confusing because the patterns in the boxes are not really distinguishable

- Neutrophils are influenced by a lot of the methods used. 1. LPS would recruit them into the site of injection. 2. Anti-L-Selectin decreases the transmigration of neutrophils into infected organs (maybe also tumors). 3. The DC-depleting mouse used has neutrophilia. All of this could influence the results by causing an oxidative burst or influencing the immune reaction. Did the authors observe neutrophil infiltration into tumors in their experimental setup? Figure 1b could be repeated with neutrophil depletion to exclude an influence of neutrophils on the NK cell reaction. Same could be said with T cells.

1st Revision - authors' response

05 May 2016

We thank this referee for his very insightful and helpful comments. The issues raised helped us to better focus the manuscript and to better introduce in the text the single experiments proposed.

The study analyzes the effect of LPS injections on the anti-tumor activity of NK cells. Tumor infiltration by lymph node-resident NK cells is enhanced by LPS injections leading to slower tumor growth. The authors observe an interaction of NK cells with dendritic cells. In addition, a stimulatory activity of IL-18 is documented.

The authors make a couple of potential interesting observations. The underlying molecular mechanisms should be worked out.

Answer: We thank the reviewer for this comment concerning the scope of the work. To better focus on the mechanism we have better introduced in the text the data previously published by our group (pages 12, 13) and we have added new original data (new figures 3, 4, 7a).

Major points:

1. *The authors do not really show that IL-18 is DC derived as they state in the abstract.*

Answer: Our previous work on DC-NK cell interaction (Zanoni I et al. Cell Rep. 2013 26:1235-49) extensively addresses the question of the origin of IL-18 necessary for NK cell activation in LPS-mediated inflammatory conditions.

In particular, starting from the observation that TLR4-deficient but not MyD88-deficient NK cells could be activated by LPS-stimulated DCs we hypothesized that either IL-1 β or IL-18 were required for the DC-mediated NK cell activation since these two cytokines require MyD88 for signaling. We therefore blocked IL-1 β and IL-18 in DC-NK cell cocultures, and we observed that only blocking IL-18 using soluble IL-18 binding protein we could block NK cell activation. We then used IL-18R-deficient NK cells and IL-18-deficient DCs and in both cases we could observe an inhibition of NK cell activation. Finally, and of great importance for this reviewer comment, we generated chimeric mice in which only DCs were deprived of the capacity to produce IL-18 in vivo. Also here we observed a significant reduction in NK cell activation in vivo following LPS administration. From all of these experiments we concluded that DC-derived IL-18 was required for NK cell activation. This point has been better introduced and clarified in the text (page 13).

Also, this manuscript extends the above-described original observation by showing that IL-18 requires contact between DCs and NK cells to be active. This is highly unusual for secreted factors, and it indicates that the local concentration of IL-18 drops sharply as the distance between DCs and NK cells increases, to the point that IL-18 concentration is no longer biologically active when DCs and NK cells are not in close proximity. Therefore, despite its soluble nature, IL-18 acts as a contact-dependent signal.

2. *The LPS is interesting but it has to shown whether this is a direct or indirect effect. On which cells does LPS act. Which genes are stimulated and which mediators are released?*

Answer: This point has also been extensively addressed in our previous work (Zanoni I et al. Cell Rep. 2013 26:1235-49) where we definitively demonstrate that LPS acts on DCs (TLR4-deficient NK cells are activated by TLR4+ DCs as efficiently as TLR4- NK cells) and not on NK cells. We also demonstrate that LPS elicits from DCs the production of IL-2, IL-18 and IFN γ , three cytokines necessary and sufficient to activate NK cells. This point has been better clarified in the text (pages 12-13).

3. *The authors treat mice with an anti-CD62L antibody to block the entry of NK cells to lymph nodes through HEVs. What else is the anti-CD62L antibody doing in vivo? This should be explained and discussed.*

Answer: The anti-L selectin CD62L antibody is commonly used in vivo to block the passage of lymphocytes through HEVs and their entry into secondary lymphoid organs. With the purpose to block NK cell entering in lymph node has been used for instance by Fontecha et al. (Martín-Fontecha AI, Thomsen LL, Brett S, Gerard C, Lipp M, Lanzavecchia A, Sallusto F Induced recruitment of NK cells to lymph nodes provides IFN-gamma for T(H)1 priming. Nat Immunol. 2004 Dec;5(12):1260-5. Epub 2004 Nov 7).

Nevertheless the possibility that the antibody has other activities in vivo cannot be totally excluded. For this reason, a second approach has been used to strengthen our results on the activation of NK cells present in the lymph node at the moment of stimulus administration.

To investigate whether LPS favors the recruitment of NK cells for their activation, CFSE-labeled NK cells have been injected iv in mice at the time of LPS injection. Adoptively transferred cells represent circulating cells that can be recruited in a preferential way at the inflamed lymph node.

Number and activation of labeled NK cells were then evaluated 5 hours after LPS administration in the draining and the contralateral lymph nodes. The prediction was that if NK cells were actively recruited by LPS at the inflamed lymph node for their activation, a larger number of CFSE+ cells should be found at the draining lymph node compared to the contralateral one. We observed that labeled NK cells injected at the time of LPS administration reached the draining and the contralateral lymph nodes with equal efficiency, suggesting homeostatic turnover rather than active recruitment (new figure 4 C). This excludes that LPS favors NK cell recruitment early after administration. Also, fractions of IFN- γ ⁺ NK cells observed in the CFSE positive and negative populations were comparable in the LPS draining lymph node (Figure 4D). This again supports the prediction that NK cells are not recruited to the inflamed lymph node to be activated, otherwise most of the CFSE+ cells should have been IFN- γ ⁺ at the peak of NK cell activation (5 hours after LPS administration). These experiments support the scenario that NK cells resident in lymph nodes at the moment of stimulus administration are the cells preferentially undergoing activation.

4. *Have the plasma levels of cytokines been measured?*

Answer: Since the stimulation with LPS is local rather than systemic, the plasma levels of cytokines would be of limited information.

5. *If IL-15 can act in an autocrine fashion, does this necessarily exclude an effect of this cytokine on NK cells? This should be explained and discussed.*

Answer: In our previous work (Zanoni I et al. *Cell Rep.* 2013 26:1235-49) we showed that both cis and trans presented IL-15 are required for optimal NK cell activation. Nevertheless NK cell activation can occur also if IL-15 is only cis- or trans-presented. Our observation has been then confirmed in human studies (Mattiola I et al *J Immunol.* 2015, 195(6):2818-28.). Since NK cells can produce and cis-present IL-15, the immune-synapse between DC and NK cells is not required to deliver the signal of IL-15 but only IL-18. This point has been clarified on page 13.

6. *What happens if the activity of IL-18 is blocked in vivo? 7. Have the authors done experiments with IL-18-/- mice? These are available from Jackson Labs.*

Answer: In our previous work (Zanoni I et al. *Cell Rep.* 2013 26:1235-49) we have generated chimeric mice in which only DCs were deprived of the ability to produce IL-18. In these mice the activation of NK cells following LPS administration was severely compromised. This has been clarified on page 13 of the revised manuscript.

Minor points:

1. *What is the molecular mechanism of FTY720? This should be explained and discussed.*

FTY720 (fingolimod) is a drug used in the treatment of multiple sclerosis (MS). FTY720 inhibits lymphocyte recirculation and induces their sequestration in the secondary lymphoid organs (M. Matloubian, C. G. Lo, G. Cinamon et al., "Lymphocyte egress from thymus and peripheral lymphoid organs is dependent on SIP receptor 1," *Nature*, vol. 427, no. 6972, pp. 355–360, 2004). Mechanistically it binds four out five sphingosine-1-phosphate "S1P" receptors, namely, S1P1, S1P3, S1P4, and S1P5; S1P1 with higher efficiency. In particular fingolimod-phosphate initially activates lymphocyte S1P1 receptor subsequently induces S1P1 down-regulation that prevents lymphocyte egress from lymphoid tissues (M. Matloubian, C. G. Lo, G. Cinamon et al., "Lymphocyte egress from thymus and peripheral lymphoid organs is dependent on SIP receptor 1," *Nature*, vol. 427, no. 6972, pp. 355–360, 2004). NK cell egress from lymph nodes depends on S1P5 and S1P1 (Jenne CNI, Enders A, Rivera R, Watson SR, Bankovich AJ, Pereira JP, Xu Y, Roots CM, Beilke JN, Banerjee A, Reiner SL, Miller SA, Weinmann AS, Goodnow CC, Lanier LL, Cyster JG, Chun J T-bet-dependent S1P5 expression in NK cells promotes egress from lymph nodes and bone marrow. *J Exp Med.* 2009 Oct 26;206(11):2469-81). Administration of high doses (like we used) of FTY720 efficiently reduces, although not completely blocks, NK cell egress from lymph nodes (K. Mayer, F. Birnbaum, T. Reinhard et al., "FTY720 prolongs clear corneal allograft survival with a differential effect on different lymphocyte populations," *British Journal of Ophthalmology*, vol. 88, no. 7, pp. 915–919, 2004; Jenne CNI, Enders A, Rivera R, Watson SR, Bankovich AJ, Pereira JP, Xu Y, Roots CM, Beilke JN, Banerjee A, Reiner SL, Miller SA, Weinmann AS, Goodnow CC, Lanier LL, Cyster JG, Chun J T-bet-dependent S1P5 expression in NK cells promotes egress from lymph nodes and bone marrow. *J Exp Med.* 2009 Oct 26;206(11):2469-81). This point has been better clarified on page 6 of the revised manuscript.

2. *Is the secretion of IL-18 stimulated via LPS on dendritic cells?*

Answer: In our previous works we extensively showed that NK cell-activating IL-18 derives from DC (Zanoni I et al. *Cell Rep.* 2013 26:1235-49) and that IL-18 is produced in very low amounts after LPS stimulation by DCs (Zanoni I et al., *Imm. Lett.* 2012 Feb 29;142(1-2):41-7). For the purpose of this study, we have generated new immunofluorescence data showing that IL-18 is secreted at the interface between DCs and NK cells in vitro. These new data have been introduced in the revised version of the manuscript (new Fig 7a).

3. *The attached movies need to be better explained.*

Answer: The legends of the attached movies are included in the supporting information.

Referee 2

We thank this referee for his very insightful and helpful comments. The issues raised helped focus our manuscript and we hope that our follow up experiments have addressed all the major concerns.

The manuscript addresses a fundamentally important question of where NK cell become activated in vivo, which is an important question for stimulating anti-tumor responses. The study proposes that lymph node NK cells and not circulating NK cells are responsible for anti-tumor responses following local administration of adjuvant, such as LPS. This concept is novel and interesting. However, I feel there are some concerns that need to be addressed to corroborate this principle.

We thank the referee for the positive comment on our work

1) *It is unclear whether all peripheral NK cells are CD62L positive and whether neutralization of CD62L entirely blocks the recruitment of peripheral NK cells into the lymph nodes. The authors should analyze CD62L KO mice and see whether NK cells are depleted in lymph nodes in steady state and after LPS injection.*

Answer: As shown in Sallusto's group paper (Martin-Fontecha A1, Thomsen LL, Brett S, Gerard C, Lipp M, Lanzavecchia A, Sallusto F *Induced recruitment of NK cells to lymph nodes provides IFN-gamma for T(H)1 priming. Nat Immunol.* 2004 Dec;5(12):1260-5.) almost all peripheral NK cells are CD62L positive. To strengthen our results on the activation of lymph node resident NK cells we have performed an additional experiment (new figure 4). CFSE-labeled NK cells were adoptively transferred at the same time of LPS administration. Number and activation of labeled NK cells were then evaluated 5 hours after LPS administration in the draining and the contralateral lymph nodes. The prediction was that if NK cells were actively recruited by LPS at the inflamed lymph node for their activation, a larger number of CFSE+ cells should have been found at the draining lymph node compared to the contralateral one. We observed that labeled NK cells injected at the time of LPS administration reached the draining and the contralateral lymph nodes with equal efficiency, suggesting regular homeostatic turnover rather than active recruitment (new figure 4 C). This excludes that LPS favors NK cell recruitment early after administration and that a preferential recruitment is necessary for NK cell activation. Fractions of IFN- γ ⁺ NK cells observed in the CFSE positive and negative populations were comparable in the LPS draining lymph node (new figure 4D). This again supports the prediction that NK cells are not recruited to the inflamed lymph node to be activated, otherwise most of the CFSE+ cells should have been IFN- γ ⁺ at the peak of NK cell activation (5 hours after LPS administration). These experiments support the scenario that lymph node resident NK cells at the moment of stimulus administration are the cells that preferentially undergo activation.

2) *Fig 3a. needs to be graphed with matched +/- FTY time points in order to evaluate whether administration of FTY influences NK cell function since it appears FTY does influence acquisition of IFN-g in contrast to the authors claims that it does not.*

Answer: The graph has been completely reorganized. Absolute numbers are shown in the new figures 3, 4.

4) *Fig 3,b I am confused as to whether egress/function are really affected here. What % of NK cells and IFNg+ NK cells are being presented? A % of total cells, or % of just NK cells? The % might be affected by a lack of leukocyte egress from the lymph node if FTY blocks. Therefore, it might be*

better to display absolute numbers of NK and IFN γ +NK; is this possible? If not, please explain.

Answer: As requested by the reviewer, absolute numbers are now shown in the new figures 3, 4. The numbers refer to one draining lymph node.

5) Fig 3c,d difficult to ascertain whether the reduced tumor vol. and %IFN-g+ NK cells in tumors is due to reduced recruitment or activation.

Answer: As shown in figure 3a the presence of FTY does not affect NK cell activation but partially affect the egress of activated cells from lymph node. We assume therefore that there is a reduced recruitment at the tumor site.

6) Fig. 4. Cells recruited to the lymph node are blocked with anti-CD62L, but the authors do not find that the % IFN γ + NK cells is significantly increased even though there is definitely an increased trend. The authors need more samples to gain statistical significance before they can conclude there is an increase otherwise they cannot claim that resident NK cells are being activated, especially since tumor vol is decreased.

Answer: Figure 4 has been significantly modified. Absolute numbers of NK cells present in the lymph node in animals treated or not with the anti-CD62L antibody are now shown. The data presented support the notion that NK cells continuously recirculate among secondary lymphoid organs and blocking NK cell entry in the lymph node for 24 hours strongly reduces the total number of NK cells present in the lymph node at the steady state. The absolute number analysis indicates that the number of activated NK cells at the draining lymph node is only minimally affected if the recruitment of blood born NK cell is blocked.

The notion that resident NK cells are activated is supported by using a second, independent approach as described above (see point 1).

7) It's easy to say lymph node resident NK are preferentially activated if you've included an anti-CD62L blocking mAb to block cellular recruitment to the lymph node. Why can't NK cells that are freshly recruited to lymph nodes also subsequently become activated? In which case what is the point of this experiment?

Answer: The recruitment, over the regular homeostatic turnover, of NK cells from blood induced by LPS takes more than 12 hours and occurs after the egress of early activated NK cells, as shown in the new figure 3b and 4a. The peak of NK cell activation is seen at 5 hours. Consistently, CD11c+ cells-derived cytokines required for NK cell activation (IL-2, IL-18 and IFN β) are produced at early time points after stimulation with LPS (from 2 to 6 hours, see Zaroni I et al. *Cell Rep.* 2013 26:1235-49). Therefore, the NK cells activated in the lymph node are preferentially those already present in the lymph node at the time of LPS administration. It is also worth noting that LPS is a transient stimulus and quickly locally inactivated by de-acylation (Lu M, Munford RS (2011) *The transport and inactivation kinetics of bacterial lipopolysaccharide influence its immunological potency in vivo.* *J Immunol* 187: 3314-3320). Therefore the effect of LPS cannot persist over time.

8) Fig 4c,d should come before c,d since this explains the above query.

Answer: there must be a typo in the request that prevented us from being able to address this point. We are open to accommodate the request once the typo is fixed.

9) Fig 4d. The experiment is proposed to show page 7 we evaluated whether NK cells could be directly activated in the skin and could reach the lymph node after activation;.

- I may be wrong, but I'm not sure there is any evaluation of whether NK cells can reach lymph nodes in this experiment because it is not clear whether the IFN-g+ NK cells present in the skin after LPS administration are resident or recruited or whether they have migrated to resident lymph nodes and that is why there is no difference in % IFN- γ + NK cells can the authors please clarify what is happening here?

Answer: we agree with the referee that this figure could be confusing; thus, we have removed it. It is very unlikely that NK cells are activated locally in the skin following LPS administration since this PAMP is quickly cleared from the skin by drainage to lymph nodes as well as local inactivation by de-acylation (Lu M, Munford RS (2011) *The transport and inactivation kinetics of bacterial lipopolysaccharide influence its immunological potency in vivo.* *J Immunol* 187: 3314-3320).

10) Figs 5 & 6 represent interesting new findings. How do the authors know that the CD11c+ cells they are analyzing are DC and not e.g. CD169+ macrophages? (Coombes et al. 2013. Cell reports) can this question be addressed to some degree?

Answer: The cells that we delete in the lymph node by diphtheria toxin administration are CD11c^{high}CD11b^{low/-} (see the image below). CD169⁺ macrophages are identified as CD11b⁺CD11c^{low}. The authors of the cited work call CD169⁺CD11c^{high} cells dendritic cells. Therefore, we are mostly depleting dendritic cells, although we cannot exclude that a few macrophages get deleted as well. As shown in the figure below, diphtheria toxin administration depletes mostly CD11c^{high} cells.

Referee 3

The techniques are great, but more statistical analysis required for the in vivo imaging, esp since the author have different conclusion versus previous study. Fairly novel in that very few studies of DC-NK interaction in vivo, but idea of NK-DC crosstalk in LN has been around for a decade. System is contrived an unlikely pertains the physiology of tumor immunity or infection. Not sure it is relevant to immune therapy.

Remarks:

In the presented manuscript, the authors provide some evidence that NK cells become activated by DCs in the lymph node to exert their antitumoral functions after stimulation with lps. The results showing a direct in vivo interaction between NK cells and DCs in the lymph node are interesting and new. The authors have modified and improved their 2-photon method, resulting in convincing data that lps induces increased interactions between adoptively transferred NK cells and DCs. The authors show nicely that the lymph node is required for endogenous NK cell activation by lps stimulation. The kinetic experiments indicate that LN activated NK cells travel to the tumor to induce antitumor immunity. The authors argue that sustained interaction between DC and NK cell in the LN leads to IL-18 induced NK cell IFN production but these data are weak.

Overall, the data are convincing and the authors interpret the data well. The finding that lps induces NK cell IFN production in the LN is not novel. In fact, Martin-Fontecha and Sallusto published these results in seminal studies in nature Immunology 2004, but as discussed in the manuscript introduction, these results are likely due to recruitment of NK cells into the LN, and not due to LN resident NK cells. The authors posit that the novelty of their findings is that resident NK cells are activated by lps in their model system. This somewhat contradicts the Sallusto findings and could be due to differences in the model system. There is a wealth of literature discussing the role of NK cell - DC crosstalk. The authors results add to this concept by showing direct in vivo interactions between NK and DC in the LN. In my opinion this is the most novel aspect of the data, although the results contradict a previous study that showed that NK-DC interactions were short-lived. The authors discuss this difference. These two apparent contradictions to the literature highlight how context-dependent the authors' results are. One way to address this issue is to use another model system, although this could be beyond the scope of the study.

Specific comments

In the methods, lps was injected sc in some exps and iv in others. This could be better defined in the figures and figure legends.

Answer: We thank the reviewer for this comment. LPS has always been injected sc. We have erroneously stated in the Mat and Met section that LPS has been injected iv. This error has been corrected.

Fig 1:

.The authors should not use "CTL" to abbreviate "control". It is used to abbreviate cytotoxic T lymphocytes.

Answer: CTL has been corrected

. Are T cells already active in the cancer model used? The protective function of LPS could also be lost in the DC-depleted mice because of loss of T cell function

Answer: We have repeated the experiment in SCID mice to address this important point, and we obtained similar results: upon LPS administration, tumor growth is reduced. This indicates that T cells are not involved. These data have been added as new Figure 1c.

Fig 2:

. In b, please correct "untreaded" into "untreated".

Answer: This typos has been corrected

. Please provide a scale bar for the microscopic pictures

Answer: The scale bar has been provided

. How many tumors were assessed?

. What is the relevance of this figure?

Answer: We have assessed three different tumors from three different mice. We observed that tumors were significantly less vascularized when NK cells had been activated. Therefore, we think that the anti-tumor effect of NK cells is most likely related to neovascularization.

Fig: 3

-interesting data

-need to discuss that effect of FTY on tumor growth could be due to blocking ag-specific T cell migration

Answer: The anti-tumor effect observed following LPS administration is T cell independent as discussed above (Figure 1c in the revised manuscript).

-cannot conclude that NK cell antitumor activity is due to their IFN production in the tumor. It is possible that NK cells make IFN to polarize to Th1 in the LN, and the T cells are required for tumor rejection.

Answer: The anti-tumor effect observed following LPS administration is T cell independent as discussed above (Fig 1C in the revised manuscript).

Fig 4:

. Anti-L-Selectin is supposed to inhibit the migration of NK cells into the lymph nodes. Here, the % of NK cells is increased. This is probably due to the antibody inhibiting the ingress of T cells, which increases the percentage of NK cells. The authors should show the data for total NK cells. Only this would show that the antibody actually inhibited the ingress of NK cells from blood into lymph nodes.

. In fact, the authors should show the effect of lps on total LN cellularity since previous studies show that LN size increased after adjuvant injection.

. Authors should show that anti-CD62L does NOT affect IFNg+ NK cell in TUMOR to confirm that ingress in LN is not important

Figure 4 has been significantly changed and absolute numbers of NK cells present in the lymph node in animals treated or not with the anti-CD62L antibody are now shown (new Figure 4a).

Total lymph node cellularity is strongly reduced in the presence of CD62L antibody (from a mean of 500,000 cells to a mean of 100,000) and does not increase following LPS administration. Absolute

NK cell numbers are also strongly reduced. The data presented support the notion that NK cells continuously recirculate through secondary lymphoid organs and blocking NK cell entry in the lymph node for 24 hours strongly reduces the total number of NK cells present in the lymph node at the steady state. The absolute number analysis indicates that the number of activated NK cells at the draining lymph node is only minimally affected if the recruitment of blood borne NK cell is blocked. See also supplementary figure 1.

To strengthen our results on the activation of lymph node resident NK cells, we followed a second, independent strategy. CFSE-labeled NK cells were adoptively transferred at the same time of LPS administration. The presence of labeled NK cells and their activation were then evaluated 5 hours after LPS administration in the draining and the contralateral lymph nodes. The prediction was that if NK cells were actively recruited by LPS at the inflamed lymph node for their activation, a larger number of CFSE+ cells should have been found at the draining lymph node compared to the contralateral one. We observed that labeled NK cells injected at the time of LPS administration reached the draining and the contralateral lymph nodes with equal efficiency, suggesting homeostatic turnover rather than active recruitment (new figure 4 C). This excludes that LPS favors NK cell recruitment early after administration in order to induce their activation. The percentage of CFSE+IFN- γ + NK cells observed in the LPS draining lymph node was comparable to that of CFSE-negative IFN- γ + (new figure 4D). This again supports the prediction that NK cells are not recruited to the inflamed lymph node to be activated, otherwise most of the CFSE+ cells should have been IFN- γ + at the peak of NK cell activation (5 hours after LPS administration). These experiments support the scenario that lymph node resident NK cells at the time of stimulus administration are the cells that preferentially undergo activation.

The presence of IFN γ + NK cells inside the tumor in mice treated with anti-CD62L has been added to the new figure 4F.

Fig 5

If I understand correctly, the authors show examples of 3 different types of DC-NK interactions after lps injection. Why is there heterogeneous interactions? Is this dependent on the timing of the lps?

Answer: The plots in this Figure show the evolution of the four parameters used in our algorithm to identify the DC-NK cells interactions. Our reasoning is that during the time course these parameters change because of the motion of the NK cells (mainly) in a random (no-interaction case) or coordinated (when there is an interaction) way. The evolution of the parameters shown in the plots is not due to the timing of LPS, which was injected 2-4 hours before these measurements. Its action should be at a regime at the time of the observations reported in Fig.4. We apologize if our explanation in the original manuscript was not sufficiently clear. The additional materials that we have now added to the revised manuscript, discussing in detail the assignment of interactions, should clarify the meaning of the plots in Fig.4.

Fig 6

-this doesn't look significant to me

-why did the authors only plot 2 parameters when 4 are used in fig 5?

-what timepoint did the author choose to represent the velocity? According to fig 5, the velocity can be drastically different (almost 5X difference in 70 min time frame)

Answer: Fig.6 indeed reports one of the major results of the manuscript, namely that the fraction of the long lasting (assumed to be > 900s) interactions increases to 12% upon LPS treatment (panel b). At the same time the average speed is not significantly changing in the two cases (panel c).

The speed reported in panel B is the average speed. The two parameters plotted in Fig.6, panels B and C, are to be considered the output of the algorithm for the detection of cell contacts. In fact the duration is the direct output of the algorithm, which is based on three parameters: instantaneous speed, cell-cell distance and confinement ratio. The speed reported in Fig.6, panel C is the average speed over the whole trajectory and is used to show that the overall motion, observed over long times (much larger than 900 s) is not sensibly affected by the interactions. The four parameters in Fig.5 are the three parameters used for the interaction algorithm and a monitor parameter $T_{on}T_{off}$. This parameter (green up-triangles) is defined as $T_{on}T_{off} = -1$ if $Dist(t_i) < d = 25 \mu m$ and $T_{on}T_{off} = -1$ if $Dist(t_i) \geq d$ and indicate the putative interactions according to a more simplified algorithm based

on the DC-NK cell distance alone. The difference in the two algorithms is remarkable and may explain at least partially some of the differences that we observe here with respect to literature results. We apologize for the lack of definition of this parameter in the original Ms, and thank the Reviewer for having pointing this out for us.

Fig 7:

- . *The title of this figure is not supported by the data.*
- . *The authors should show that LPS-stimulated DCs indeed secrete IL-18 in their hands. If they don't show that, they cannot say that these cells are insufficient to induce IFN release by NK cells. Maybe the release of IL-18 by DCs did just not work in their experiment.*
- . *This experiment is insufficient to make the conclusion "IL-18 is the contact-dependent signal required in DC-NK cell interactions". So far, the authors have only shown that it is sufficient. The contact-dependence is in no way shown, b suggests both (NK cells can still be activated even though they don't come in direct contact with DCs, as long as DCs are stimulated with LPS). The authors should perform additional experiments.*
- . *In fact, I would interpret the data to show that lps+DC induces a soluble factor that can activate NK cell IFN γ in the presence of rIL-18 given to the NK cell.*
- . *The dependence on IL-18 is not shown. DCs deficient in IL-18 would be the definitive experiment here.*
- . *Title should be changed or more experiments done.*

Answer: In our previous work (Zanoni I et al. *Cell Rep.* 2013 26:1235-49) we have extensively shown that DC-derived IL-18 is required for NK cell activation.

In particular, starting from the observation that TLR4-deficient but not MyD88-deficient NK cells could be activated by LPS-stimulated DCs we hypothesized that either IL-1b or IL-18 were required for the DC-mediated NK cell activation since these two cytokines require MyD88 for signaling. We therefore blocked IL-1b and IL-18 in DC-NK cell cocultures, and we observed that only blocking IL-18 using soluble IL-18 binding protein we could block NK cell activation. We then used IL-18R-deficient NK cells and IL-18-deficient DCs and in both cases we could observe an inhibition of NK cell activation. Finally we generated chimeric mice in which only DCs were deprived of the capacity to produce IL-18 in vivo and again we could observe a significant reduction in NK cell activation in vivo following LPS administration. From all of these experiments we concluded that DC-derived IL-18 was required for NK cell activation.

The other two DC-derived cytokines we have found indispensable for NK cell activation in the presence of LPS are IL-2 and IFN β . IFN β , in turn, is required to elicit IL-15 and IL-15Ra production from both DCs and NK cells. IL-15 induces NK cell activation via cis- and trans-presentation.

This point has been better clarified in the revised version of the manuscript (page 12). In the present manuscript we have added an experiment showing that IL-18 is secreted at the contact site between DC-NK cells (new figure 7A).

Minor comments

The legends in Figures 3c and 4c are confusing because the patterns in the boxes are not really distinguishable

Answer: new figures 3c and 4c have been introduced in the revised version of the manuscript.

Neutrophils are influenced by a lot of the methods used. 1. LPS would recruit them into the site of injection. 2. Anti-L-Selectin decreases the transmigration of neutrophils into infected organs (maybe also tumors). 3. The DC-depleting mouse used has neutrophilia. All of this could influence the results by causing an oxidative burst or influencing the immune reaction. Did the authors observe neutrophil infiltration into tumors in their experimental setup? Figure 1b could be repeated with neutrophil depletion to exclude an influence of neutrophils on the NK cell reaction. Same could be said with T cells.

The experiment has been repeated in SCID mice and the same reduction in tumor growth has been observed (new figure 1B), indicating that T cells are not involved. Concerning neutrophils, it is clear that eliminating NK cells with the anti-asialoGM antibody completely abolished the effect on tumor growth (Figure 1B), indicating that NK cells are responsible for this phenomenon. Also, in our

previous work (Zanoni *et al* JCI 2012) we have shown that sc LPS administration, at the same conditions used here, does not induce neutrophil recruitment at the site of injection. Nevertheless, we have analyzed the presence of neutrophils inside the tumor. We have observed that, after LPS treatment, the presence of neutrophils inside the tumor is strongly reduced; this data are included here for the referee perusal. This observation suggests that activated NK cells may exert their anti-tumor activity not only in a direct way via the production of IFN- γ and its anti-angiogenic function, but also in an indirect way by reducing the ingress of neutrophils inside the tumor. There is evidence in literature of an inhibitory effect of NK cell-derived IFN- γ on neutrophil recruitment at the inflammation site in autoimmune and infection-mediated inflammatory conditions. For instance, in an inflammatory arthritis model, the arrival of neutrophils in the joints is blocked by NK-derived IFN- γ (Wu H. J., Sawaya H., Binstadt B., Brickelmaier M., Blasius A., Gorelik L., Mahmood U., Weissleder R., Carulli J., Benoist C., Mathis D. (2007) *Inflammatory arthritis can be reined in by CpG-induced DC-NK cell cross talk*. *J. Exp. Med.* 204, 1911–1922). In a model of M. tuberculosis infection NK cell-derived IFN- γ negatively regulates the lung infiltration of neutrophils by modulating the levels of neutrophil chemotactic molecules (Feng C. G., Kaviratne M., Rothfuchs A. G., Cheever A., Hieny S., Young H. A., Wynn T. A., Sher A. (2006) *NK cell-derived IFN- γ differentially regulates innate resistance and neutrophil response in T cell-deficient hosts infected with Mycobacterium tuberculosis*. *J. Immunol.* 177, 7086–7093). Finally in an endotoxin-induced uveitis model the production of IFN- γ by NK1.1+ cells, reduced levels of the chemokine KC and neutrophil infiltration (Figueiredo F., Commodaro A. G., de Camargo M. M., Rizzo L. V., Belfort R. Jr. (2007) *NK1.1 cells downregulate murine endotoxin-induced uveitis following intraocular administration of interleukin-12*. *Scand. J. Immunol.* 66, 329–334). This point has been discussed on pag 14 of the revised manuscript-

The identification of the mechanisms through which NK cells-derived IFN- γ exerts anti-tumor functions is beyond the scope of this work. It is, nevertheless, a very interesting point and we will devote further attention to it.

2nd Editorial Decision

01 June 2016

Thank you for the submission of your revised manuscript to EMBO Molecular Medicine. We have now received the enclosed reports from the referees that were asked to re-assess it. As you will see the reviewers are now globally supportive and I am pleased to inform you that we will be able to accept your manuscript pending the following final amendments:

- 1) Reviewer 3 would like you to clarify a few remaining points. Please deal with these appropriately, especially with respect to the question on significance. Provided you do so carefully, I will make an editorial decision on your next final version.
- 2) As per our Author Guidelines, the description of all reported data that includes statistical testing must state the name of the statistical test used to generate error bars and P values, the number (n) of independent experiments underlying each data point (not replicate measures of one sample), and the actual P value for each test (not merely 'significant' or 'P < 0.05').
- 3) We encourage the publication of source data, particularly for electrophoretic gels and blots, with the aim of making primary data more accessible and transparent to the reader. Would you be willing

to provide a PDF file per figure that contains the original, uncropped and unprocessed scans of all or at least the key gels used in the manuscript? The PDF files should be labeled with the appropriate figure/panel number, and should have molecular weight markers; further annotation may be useful but is not essential. The PDF files will be published online with the article as supplementary "Source Data" files. If you have any questions regarding this just contact me.

4) The manuscript must include a statement in the Materials and Methods identifying the institutional and/or licensing committee approving the experiments, including any relevant details (like how many animals were used, of which gender, at what age, which strains, if genetically modified, on which background, housing details, etc). We encourage authors to follow the ARRIVE guidelines for reporting studies involving animals. Please see the EQUATOR website for details: <http://www.equator-network.org/reporting-guidelines/improving-bioscience-research-reporting-the-arrive-guidelines-for-reporting-animal-research/>. I note that you have provided most details in the author checklist, but it is also important that you also integrate the manuscript with additional information.

5) Please adjust the figure callouts in the article according to our guidelines (<http://embomolmed.embopress.org/authorguide>) and refer to movies or videos consistently and correctly. Furthermore, the appendix file is missing the Table of Contents and features some erroneous figure labels; it also needs to be submitted as a PDF file.

6) The movie legends need to be removed from the Appendix file. Please provide each movie legend as a read-me file uploaded together with the movie-file as zip file - per each movie please!

7) Please remove all red lettering from the manuscript and the appendix as it is no longer needed.

8) I have slightly edited the Abstract section (please see attached manuscript) and suggested an alternative title. Please accept/modify as appropriate using the attached version. I should add that I find the parts of the "problem" paragraph in your "The Paper Explained" section to be more compelling and efficacious than the Abstract introductory part. You might consider modifying the Abstract a little in that respect.

Please submit your revised manuscript within two weeks. I look forward to seeing a revised form of your manuscript as soon as possible.

***** Reviewer's comments *****

Referee #1 (Comments on Novelty/Model System):

The authors have convinced with their revision

Referee #1 (Remarks):

The study analyzes the effect of LPS injections on the anti-tumor activity of NK cells. Tumor infiltration by lymph node-resident NK cells is enhanced by LPS injections leading to slower tumor growth. The authors observe an interaction of NK cells with dendritic cells. In addition, a stimulatory activity of IL-18 is documented.

This is a very interesting study and the authors have now satisfactorily addressed all my concerns.

Referee #2 (Remarks):

The authors have thoroughly addressed my concerns and I think the study now provides a significant advance that deserves publication in EMBO molecular Medicine

Referee #3 (Comments on Novelty/Model System):

This is the only feasible model system to study recirculating NK cells. The authors do a good job of demonstrating regulation of NK cell activation by lps in the context of recirculating NK cells in the LN.

Referee #3 (Remarks):

This manuscript has been improved and now shows nicely that NK cells recirculate (interesting finding) and that LN resident NK cells get activated by LPS via DC interactions.

Minor comments:

1. Fig 1C should be same style (line thickness and color) as other figs.
2. Fig 2b "untreated" needs to be changed (pointed out in first review, but still not done!)
3. Disagree with interp of Fig 4B: in fact, the number of activated NK cells in the draining lymph nodes is strongly reduced (about half) after CD62L antibody treatment, indicating that the migration of circulating NK cells into the LNs does have some effect. Are these results significant? They look significant but the stars were left out.

2nd Revision - authors' response

09 June 2016

We are pleased that the reviewers were satisfied with our revision. We have edited the manuscript and figures to further clarify any possible remaining confusing point and to meet your suggestions. In particular:

- 1) *Reviewer 3 would like you to clarify a few remaining points. Please deal with these appropriately, especially with respect to the question on significance. Provided you do so carefully, I will make an editorial decision on your next final version.*

Referee #3 (Remarks):

This manuscript has been improved and now shows nicely that NK cells recirculate (interesting finding) and that LN resident NK cells get activated by LPS via DC interactions.

Minor comments:

1. *Fig 1C should be same style (line thickness and color) as other figs.*

Style of Fig 1C has been changed.

2. *Fig 2b "untreated" needs to be changed (pointed out in first review, but still not done!)*

Untreated has been removed and substituted with NT, similarly to other figures.

3. *Disagree with interp of Fig 4B: in fact, the number of activated NK cells in the draining lymph nodes is strongly reduced (about half) after CD62L antibody treatment, indicating that the migration of circulating NK cells into the LNs does have some effect. Are these results significant? They look significant but the stars were left out.*

We thank the reviewer for this comment that helped us to further clarify this important point of the manuscript. Mice treated with anti-CD62L have a strongly reduced number of both total NK cells and IFN- γ + NK cells in steady state conditions. When mice are treated with LPS the absolute numbers of NK cells that become IFN- γ + are very similar in mice treated or not with anti-CD62L (as depicted in the figure below).

New IFN γ + NK cells

These numbers are calculated by subtracting to the total number of IFN- γ + NK cells the numbers of basally activated cells. Therefore, although the total number of activated cells appears to be higher

in mice that did not receive the anti-CD62L antibody, this is only due to the higher number of basally activated cells. The numbers of newly activated cells after LPS treatment is only minimally affected by the antibody treatment (see figure). This minimal, non statistically significant, difference in the numbers of newly activated cells is likely due to the regular homeostatic recirculation that is not present in anti-CD62L treated mice.

We have introduced a further clarification in the text to stress this point (page 7, lines 3-10).

- 2) *As per our Author Guidelines, the description of all reported data that includes statistical testing must state the name of the statistical test used to generate error bars and P values, the number (n) of independent experiments underlying each data point (not replicate measures of one sample), and the actual P value for each test (not merely 'significant' or 'P < 0.05').*

The actual p values have been added in each figure and the numbers of mice used added in each legend. All the statistical analyses have been performed according to the paragraph "Statistical analysis" in the Material and Methods section.

- 3) *We encourage the publication of source data, particularly for electrophoretic gels and blots, with the aim of making primary data more accessible and transparent to the reader. Would you be willing to provide a PDF file per figure that contains the original, uncropped and unprocessed scans of all or at least the key gels used in the manuscript? The PDF files should be labeled with the appropriate figure/panel number, and should have molecular weight markers; further annotation may be useful but is not essential. The PDF files will be published online with the article as supplementary "Source Data" files. If you have any questions regarding this just contact me.*

Electrophoretic gels and blots are not present in the manuscript.

- 4) *The manuscript must include a statement in the Materials and Methods identifying the institutional and/or licensing committee approving the experiments, including any relevant details (like how many animals were used, of which gender, at what age, which strains, if genetically modified, on which background, housing details, etc). We encourage authors to follow the ARRIVE guidelines for reporting studies involving animals. Please see the EQUATOR website for details: <http://www.equator-network.org/reporting-guidelines/improving-bioscience-research-reporting-the-arrive-guidelines-for-reporting-animal-research/>. I note that you have provided most details in the author checklist, but it is also important that you also integrate the manuscript with additional information.*

The required additional information concerning the studies involving mice has been added in the Material and Methods section (*Mice paragraph*).

- 5) *Please adjust the figure callouts in the article according to our guidelines (<http://embomolmed.embopress.org/authorguide>) and refer to movies or videos consistently and correctly. Furthermore, the appendix file is missing the Table of Contents and features some erroneous figure labels; it also needs to be submitted as a PDF file.*

The figure callout has been changed. The Table of Contents has been added to the appendix file and figure labels have been corrected.

- 6) *The movie legends need to be removed from the Appendix file. Please provide each movie legend as a read-me file uploaded together with the movie-file as zip file - per each movie please!*

A zip file has been generated for each movie including the movie and its legend.

- 7) *Please remove all red lettering from the manuscript and the appendix as it is no longer*

needed.

Red lettering has been removed.

- 8) *I have slightly edited the Abstract section (please see attached manuscript) and suggested an alternative title. Please accept/modify as appropriate using the attached version. I should add that I find the parts of the "problem" paragraph in your "The Paper Explained" section to be more compelling and efficacious than the Abstract introductory part. You might consider modifying the Abstract a little in that respect.*

Title and abstract have been slightly changed according to your suggestions.

Corresponding author name: Ivan Zanoni, Francesca Granucci

Manuscript Number: EMM-2015-06163